# A non canonical subtilase attenuates the transcriptional activation of defence responses in *Arabidopsis thaliana*

Irene Serrano[1†], Pierre Buscaill[1†], Corinne Audran[1], Cécile Pouzet[2], Alain Jauneau[2], Susana Rivas[1*]

[1]LIPM, Université de Toulouse, INRA, CNRS, Castanet-Tolosan, France; [2]Fédération de Recherche 3450, Plateforme Imagerie, Pôle de Biotechnologie Végétale, Castanet-Tolosan, France

**Abstract** Proteases play crucial physiological functions in all organisms by controlling the lifetime of proteins. Here, we identified an atypical protease of the subtilase family [SBT5.2(b)] that attenuates the transcriptional activation of plant defence independently of its protease activity. The *SBT5.2* gene produces two distinct transcripts encoding a canonical secreted subtilase [SBT5.2(a)] and an intracellular protein [SBT5.2(b)]. Concomitant to *SBT5.2(a)* downregulation, *SBT5.2(b)* expression is induced after bacterial inoculation. SBT5.2(b) localizes to endosomes where it interacts with and retains the defence-related transcription factor MYB30. Nuclear exclusion of MYB30 results in its reduced transcriptional activation and, thus, suppressed resistance. *sbt5.2* mutants, with abolished *SBT5.2(a)* and *SBT5.2(b)* expression, display enhanced defence that is suppressed in a *myb30* mutant background. Moreover, overexpression of SBT5.2(b), but not SBT5.2(a), in *sbt5.2* plants reverts the phenotypes displayed by *sbt5.2* mutants. Overall, we uncover a regulatory mode of the transcriptional activation of defence responses previously undescribed in eukaryotes.

**\*For correspondence:** susana. rivas@toulouse.inra.fr

[†]These authors contributed equally to this work

**Competing interests:** The authors declare that no competing interests exist.

## Introduction

Regulation of protein turnover plays a central role in the proper functioning of eukaryotic cells. Indeed, proteases of different families have been shown to be involved in the control of metabolism, physiology, growth and adaptive responses to biotic and abiotic stimuli (*van der Hoorn, 2008*). Subtilisin-like proteases (subtilases) are serine proteases featuring a catalytic triad characterized by the three amino acids aspartate, histidine and serine (*Dodson and Wlodawer, 1998*). According to the MEROPS (http://merops.sanger.ac.uk) classification, subtilases belong to the S8 family within the SB clan of serine proteases and are grouped into two subfamilies, subtilisins (S8A) and kexins (S8B). Nine subtilases have been characterized in mammals, seven of which belong to the kexin group and two others to the S8A subfamily (pyrolysins). With no representatives of the kexin type, plant subtilases exclusively belong to the pyrolysin group within the S8A subfamily. Pyrolysin-related subtilase families are largely expanded throughout the plant kingdom with a degree of complexity that exceeds that of their mammalian counterparts (*Schaller et al., 2012*). In *Arabidopsis* the subtilase family comprises 56 members distributed in six distinct subgroups (SBT1-6) (*Rautengarten et al., 2005*). Despite their prevalence, our knowledge of the function of plant subtilases is rather poor. Subtilases are predicted to be secreted and have been involved in general protein turnover as well as in the highly specific regulation of plant development or responses to environmental changes and, more recently, in suppression of basal immunity and immune priming (*Schaller et al., 2012*; *Figueiredo et al., 2014*).

**eLife digest** Like animals, plants have evolved numerous ways to protect themselves from disease. When a plant detects an invading microbe, it massively changes which genes it expresses to establish a defensive response. This is possible thanks to the action of a type of protein, named transcription factors, which are able to bind to DNA in the cell nucleus and regulate gene expression. However, triggering such a response comes at a cost, and so plants must keep their defensive response in check such that they can allocate resources in a balanced way.

In the model plant *Arabidopsis*, a protein named MYB30 is one transcription factor that is able to promote disease resistance. Previous research identified some proteins that can reduce the activity of this transcription factor to avoid triggering a response when it is not needed, for example, when no infectious microbes are present. However, it was likely that other proteins were also involved in the process.

Now, Serrano et al. report that an enzyme called SBT5.2 is an additional negative regulator of MYB30 activity. SBT5.2 belongs to a family of protein-degrading enzymes called subtilases, which are typically localized outside cells. As such, it was unclear how SBT5.2 could interact and regulate a transcription factor that is found inside the nucleus of plant cells.

Nevertheless, Serrano et al. found that the gene that encodes SBT5.2 actually gives rise to two distinct proteins. The first is a classical subtilase that is indeed located outside of the cell, and so cannot interact with MYB30 and does not affect its activity. The second protein is an atypical subtilase that localises to bubble-like compartments called vesicles within the cell and is able to highjack MYB30 on its way to the nucleus. When the atypical subtilase interacts with MYB30 at vesicles, it stops MYB30 from entering the nucleus. As a result, MYB30 cannot bind to the DNA nor activate its target genes. This means that the defensive response that normally depends on MYB30 is weakened.

The work of Serrano et al. uncovers a new way to regulate the expression of defence-related genes. Further unravelling the molecular mechanisms involved in the fine-tuning of gene expression represents a challenging task for future research.

As sessile organisms, plants must face the diversity of pathogens that they encounter in their habitat. Plants, unlike mammals, lack mobile defender cells and a somatic adaptive immune system. Instead, they rely on the innate immunity of each cell and on systemic signals originating from infection sites. Plant resistance to disease is a costly response, closely connected to plant physiological and developmental processes, and often associated with the so-called hypersensitive response (HR), a form of programmed cell death that develops at attempted infection sites, allegedly to prevent pathogen propagation through the plant (*Coll et al., 2011*). The sharp limit of the HR suggests the existence of tight regulatory mechanisms to restrict cell death development to the inoculated zone although the molecular actors involved in this process remain unknown for the most part. In line with the high cellular cost of triggering defence and cell death-associated responses, negative regulatory mechanisms are used by the plant to attenuate the activation of immune-related functions and allow a balanced allocation of resources upon pathogen challenge.

Transcriptional reprogramming of the plant cell is a crucial step that allows mounting of efficient defence responses after pathogen attack. Transcription factors (TFs) and co-regulatory proteins play essential roles in launching and regulating the transcriptional changes that direct the plant defence response (*Buscaill and Rivas, 2014*; *Tsuda and Somssich, 2015*). MYB TFs of the R2R3 type (126 members in *Arabidopsis*) mostly regulate plant-specific functions (*Dubos et al., 2010*). Among the MYB TFs regulating defence-related transcription, *Arabidopsis* MYB30 is one of the best characterized. MYB30 promotes defence and cell death-associated responses through the transcriptional activation of genes related to the lipid biosynthesis pathway that leads to the production of very-long-chain fatty acids (VLCFAs) (*Raffaele et al., 2008*). MYB30 is targeted by the effector protein XopD, from the bacterial pathogen *Xanthomonas campestris* pv. *campestris (Xcc)*, leading to suppression of MYB30-mediated transcriptional activation of plant resistance and thus underlining the important role played by MYB30 in plant defence regulation (*Canonne et al., 2011*).

A previously performed yeast two-hybrid (Y2H) screen using MYB30 as bait identified a secretory phospholipase (*AtsPLA₂-α*) and a RING-type E3 ubiquitin ligase (MIEL1) that exert negative spatial and temporal control on MYB30 transcriptional activity through distinct molecular mechanisms (*Froidure et al., 2010*; *Marino et al., 2013*). Here, we describe SBT5.2, a serine protease of the subtilisin group, as a new MYB30-interacting partner. We demonstrate that the *SBT5.2* transcript is alternatively spliced and that one of the two *SBT5.2* splice variants, SBT5.2(b), whose expression pattern follows that of *MYB30* after bacterial treatment, encodes an atypical subtilase that specifically mediates retention of MYB30 at endosomal vesicles. This phenomenon is independent of the integrity of the SBT5.2(b) catalytic triad, requires N-terminal myristoylation of SBT5.2(b) and results in attenuation of MYB30-mediated HR. Our work uncovers a novel regulatory mode for a subtilase protein and underlines the intricacy of the transcriptional regulation of plant responses to pathogen attack.

## Results

### Identification of SBT5.2

In order to search for components involved in MYB30-mediated signalling, a Y2H screen was previously conducted using a MYB30 version deleted from its transcriptional activation domain (MYB30ΔAD) (*Froidure et al., 2010*) as bait. A cDNA clone encoding the last 103 amino acids of the Arabidopsis serine protease of the subtilisin group SBT5.2 (At1g20160) was identified in this screen (*Figure 1*). SBT5.2 belongs to subgroup V (6 members) within the classification of the Arabidopsis subtilase family (*Schaller et al., 2012*; *Rautengarten et al., 2005*).

Two gene models are annotated in the TAIR database (http://www.arabidopsis.org) for *At1g20160*, suggesting that the corresponding transcript is alternatively spliced (*Figure 2a*). Comparison of the two *SBT5.2* cDNA clones [designated *SBT5.2(a)* and *SBT5.2(b)*] with the genomic sequence revealed that the first intron is specifically spliced in the *SBT5.2(b)* cDNA (*Figure 2—figure supplement 1a*). Reverse transcription PCR (RT-PCR) using specific cDNA primers confirmed the existence of both splice variants *in planta* (*Figure 2b*). The sequence of *SBT5.2(a)* and *SBT5.2(b)* 5' ends was determined by 5' RACE and cDNA sequencing (*Figure 2—figure supplement 1b*). To gain knowledge on the relative abundance of both isoforms after bacterial inoculation, we monitored the expression of *SBT5.2(a)* and *SBT5.2(b)* in Col-0 wild-type plants inoculated with *Pseudomonas syringae* pv. *tomato* DC3000 expressing the avirulence gene *AvrRpm1 (Pst AvrRpm1)*. As shown in *Figure 2c*, expression of *SBT5.2(a)* was downregulated after treatment with bacteria, whereas

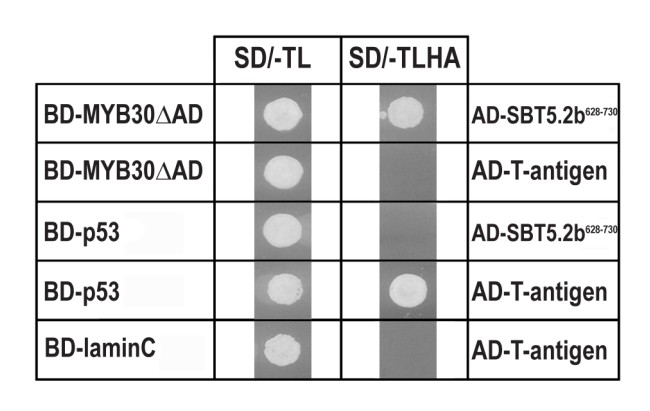

**Figure 1.** Specific interaction between MYB30 and SBT5.2 in yeast. Yeasts are shown after growth for five days on low stringency (left; SD/-TL) or high stringency (right; SD/-TLHA) media. Co-expression of MYB30 deleted from its C-terminal transcription activation domain (MYB30△AD) and the isolated cDNA clone encoding the last 103 amino acids of SBT5.2 (SBT5.2$^{628-730}$) resulted in yeast growth on selective medium. In a control experiment, yeast cells expressing MYB30△AD or SBT5.2$^{628-730}$ with controls provided by Clontech (T-antigen or P53, respectively) were not able to grow on selective medium. BD, GAL4 DNA-binding domain; AD, GAL4 activation domain.

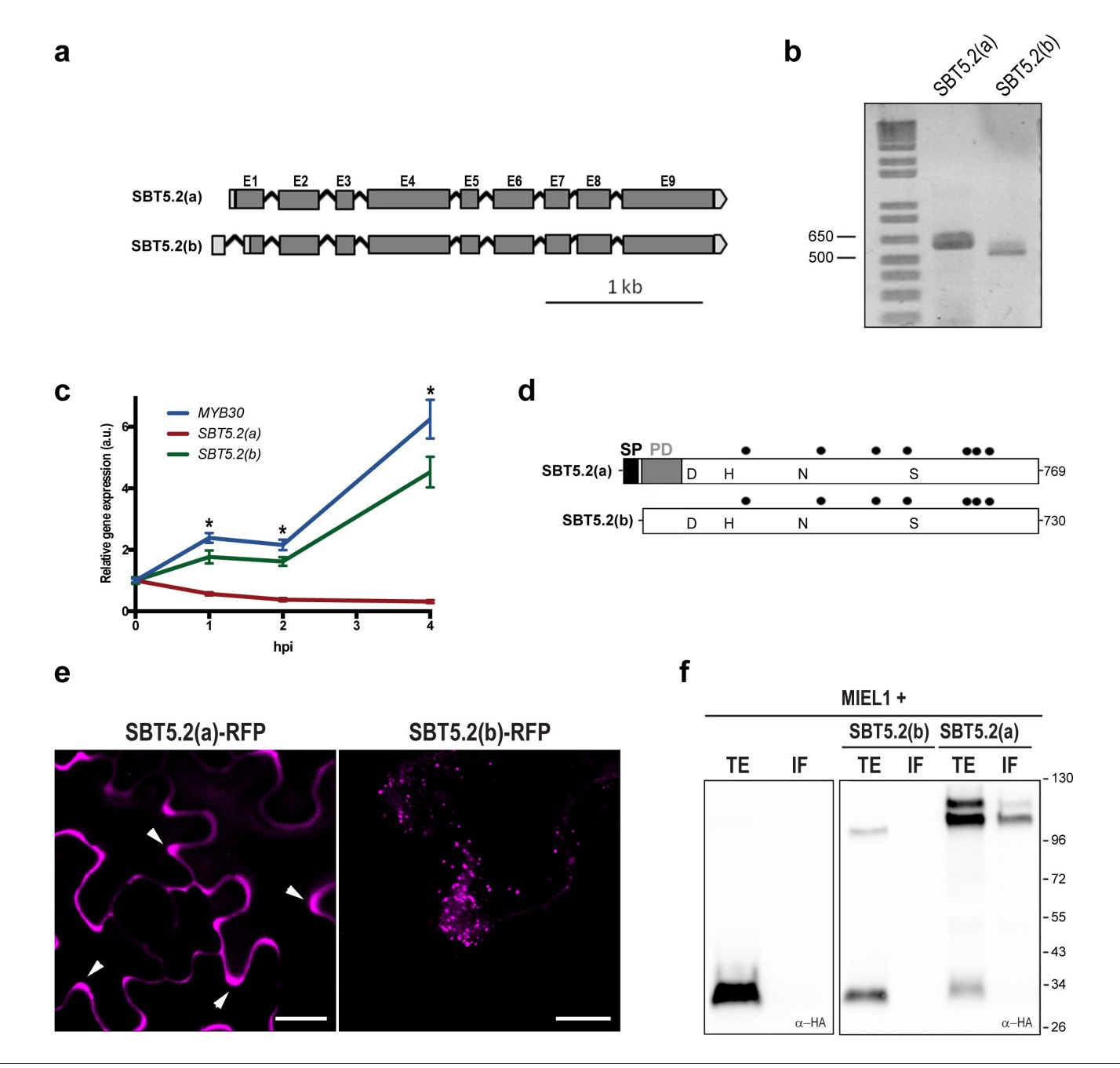

**Figure 2.** SBT5.2 is alternatively spliced. (**a**) Genomic structure of *SBT5.2* alternatively spliced variants. Exons are shown as dark gray boxes (E1-E9), introns as black lines between exons and 5' and 3' UTRs are shown in light gray. (**b**) RT-PCR analysis of *SBT5.2(a)* and *SBT5.2(b)* transcripts in four-week old Col-0 Arabidopsis leaves. (**c**) The relative expression of *SBT5.2(a), SBT5.2(b)* and *MYB30* at the indicated timepoints after inoculation of Col-0 plants with *Pst AvrRpm1* ($5 \times 10^7$ cfu/ml). Expression values were normalized using *SAND* family gene as internal standard and related to the value of each gene at time 0, which is set at 1. The SEM values were calculated from 4 independent experiments (4 replicates/experiment). The asterisks indicate statistically significant values for the three tested genes according to a Student's *t*-test (p<0.005) and with respect to gene expression values at time 0. (**d**) Schematic representation of SBT5.2(a) and SBT5.2(b) protein sequences. The signal peptide (SP) and the pro-domain (PD) in the SBT5.2(a) isoform are shown as black and grey boxes, respectively. Catalytically conserved Asp, His, Asn, and Ser residues are indicated. Putative N-glycosylation sites (PGSs) are indicated by black dots. (**e**) Confocal images of epidermal *N. benthamiana* cells 36 hr after *Agrobacterium*-mediated transient expression of the indicated constructs. Accumulation of SBT5.2(a)-RFP in apoplastic spaces is indicated by arrowheads. Bars = 10 μm. (**f**) Western blot analysis of total protein extracts (TE) and intercellular fluids (IF) from *N. benthamiana* leaves expressing the intracellular protein MIEL1 alone (left) or co-expressed with HA-tagged SBT proteins (right), as indicated. Molecular mass markers in kilodaltons are indicated on the right.

*Figure 2 continued on next page*

*Figure 2 continued*

The following figure supplement is available for figure 2:

**Figure supplement 1.** Alternative splicing of *SBT5.2* gives raise to two distinct variants.

expression of *SBT5.2(b)* was induced and displayed an expression profile highly similar to that of *MYB30* (*Figure 2c*). This data suggests that SBT5.2(b) may have a MYB30-related function.

*SBT5.2(a)* corresponds to a transcript of 2402 bp, which is predicted to encode a 769 amino acid preproenzyme containing a signal peptide (SP) followed a by prodomain (PD), which acts as an intra-molecular inhibitor, and a mature polypeptide (*Figure 2d*; *Figure 2—figure supplement 1c*). In contrast, *SBT5.2(b)* corresponds to a transcript of 2373 bp, predicted to encode a protein of 730 amino acids with no SP and lacking the first five amino acids of the PD (*Figure 2d*; *Figure 2—figure supplement 1c*). Except for their N-terminal differences, the two corresponding encoded proteins are predicted to be identical and contain a catalytic triad with three amino acids (D145, H210 and S546 in SBT5.2(a), and D106, H171 and S507 in SBT5.2(b)) conserved within serine proteases (*Figure 2d*; *Figure 2—figure supplement 1b*).

## SBT5.2(a) is a secreted protein whereas SBT5.2(b) is intracellular

Alternative splicing (AS) of *SBT5.2* may have important implications for the subcellular localization and function of the proteins encoded by the two transcripts. The presence of a SP and a PD in SBT5.2(a) suggests that this protein may enter the secretory pathway and be secreted to the extracellular space. Indeed, secretion of SBT5.2(a) was previously reported (*Engineer et al., 2014*; *Kaschani et al., 2012*). In contrast, the absence of the SP in SBT5.2(b) may prevent secretion of the protein. In order to test this possibility, the subcellular localization of the two proteins was first investigated using *Agrobacterium*-mediated transient expression of RFP-tagged SBT5.2(a) and SBT5.2(b) under the control of a dexamethasone- (Dex-) inducible promoter in leaf epidermal cells of *N. benthamiana*. As expected, SBT5.2(a) was found to be located in apoplastic spaces whereas SBT5.2(b) was detected at vesicle-like structures inside cells (*Figure 2e*).

To obtain biochemical validation of the subcellular localization of SBT5.2(a) and SBT5.2(b), HA-tagged versions of both proteins were transiently expressed in *N. benthamiana* and intercellular fluid (IF) was isolated. In order to control the detection of intracellular proteins in the IF, the intracellular protein MIEL1 (*Marino et al., 2013*) was co-expressed with SBT5.2 proteins. As expected for an intracellular protein, MIEL1 was detected in the total extract fraction (TE) and not in the IF, confirming that the IF fraction did not contain intracellular proteins due to unintentional cellular lysis during IF isolation (*Figure 2f*). SBT5.2(a) was detected in the IF as two protein bands that may correspond to the processed and unprocessed forms of the protease, whereas SBT5.2(b) was exclusively detected in the TE and never in the IF fraction (*Figure 2f*). These results confirm the secretion of SBT5.2(a) and the intracellular localization of SBT5.2(b).

## SBT5.2(a), but not SBT5.2(b), is N-glycosylated

Most extracellular or secreted proteins are modified via N-glycosylation (*Moremen et al., 2012*). Seven putative *N*-linked glycosylation sites (PGSs; N in NxS/T motifs) are present in SBT5.2 proteins [N225, N363, N467, N525, N636, N650 and N678 in SBT5.2(a)]. In order to test whether SBT5.2 proteins are glycosylated *in planta*, protein extracts containing SBT5.2(a) and SBT5.2(b) were treated with PNGase F or EndoH and analysed for mobility shifts by Western Blot. At the end of their maturation in the secretory pathway, some plant N-linked glycans are modified, which renders them resistant to cleavage by glycosylases (*Lerouge et al., 1998*). In agreement, an electrophoretic shift was only observed for the slow migrating, unprocessed form of SBT5.2(a), whereas migration of the fully processed form remained unaltered (*Figure 3a*). In addition, SBT5.2(a) bound to and was eluted from a concanavalin A resin (*Figure 3b*), further suggesting that SBT5.2(a) is a glycosylated protein. Finally, the increased electrophoretic mobility of SBT5.2(a) in protein extracts from *N. benthamiana* leaves expressing HA-tagged SBT5.2(a) and treated with tunicamycin, an inhibitor of *N*-linked

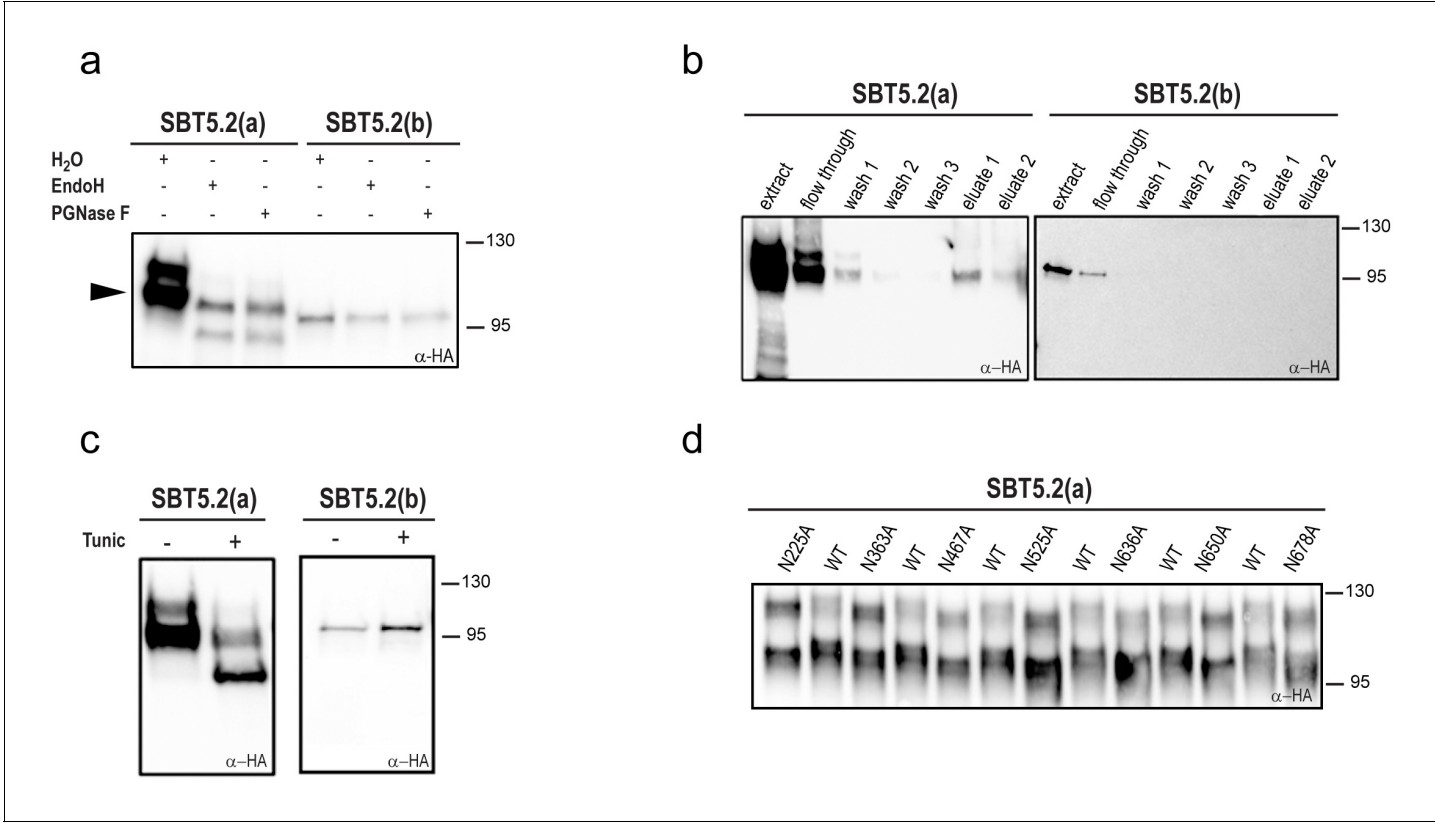

**Figure 3.** SBT5.2(a), but not SBT5.2(b), is glycosylated *in planta*. (**a**) SBT5.2(a), but not SBT5.2(b), is deglycosylated by PNGase F and Endo H. Protein extracts containing HA-tagged SBT5.2(a) and SBT5.2(b) transiently expressed in *N. benthamiana* were treated (+) or not (−) with PNGase F or Endo H as indicated. The arrowhead on the left indicates the fully processed form of SBT5.2(a) whose mobility is not affected by the enzymatic treatment. (**b**) HA-tagged SBT5.2(a), but not SBT5.2(b) can be affinity purified using a concanavalin A resin. (**c**) Glycosylation of HA-tagged SBT5.2(a), but not SBT5.2 (b), is blocked by tunicamycin treatment (+) after transient expression in *N. benthamiana*. (**d**) Electrophoretic mobility of individual HA-tagged PGS SBT5.2(a) mutants. Mutated N to A residues are indicated. WT: wild-type SBT5.2(a) proteins were interspersed to facilitate detection of the mobility shifts. In all cases, Western blot analyses were performed using anti-HA antibodies.

glycosylation of newly synthesized glycoproteins in the ER (*Bassik and Kampmann, 2011*), further confirmed SBT5.2(a) N-glycosylation *in planta* (*Figure 3c*).

We next analyzed individual PGS removal mutants (in which the N residue was replaced by A) on high resolution SDS-PAGE gels to determine their electrophoretic mobility as compared to wild-type SBT5.2(a). This analysis revealed a small but significant mobility shift for all SBT5.2(a) PGS mutants (*Figure 3d*), suggesting that all PGS in SBT5.2(a) are used *in planta*.

Despite the fact that SBT5.2(b) contains the seven PGSs present in SBT5.2(a), (i) SBT5.2(b) electrophoretic mobility was not modified after treatment with PNGase F or EndoH (*Figure 3a*); (ii) neither SBT5.2(b) binding to nor elution from a concanavalin A resin was observed (*Figure 3b*); and (iii) no effect of tunicamycin leaf treatment on the SBT5.2(b) electrophoresis profile was detected (*Figure 3c*). These results, which are consistent with the absence of SP in SBT5.2(b) and our previous observation that SBT5.2(b) is not secreted, strongly suggest that SBT5.2(b) does not enter the secretory pathway and is therefore not *N*-glycosylated.

## SBT5.2(a), but not SBT5.2(b), shows serine protease activity

Subtilases, as other proteases, are typically able to catalyze their self-processing to render a mature active polypeptide. The presence of the three conserved amino acids in the catalytic triad of SBT5.2 proteins is consistent with these proteins displaying protease activity. When transiently expressing SBT5.2(a) in leaf epidermal cells of *N. benthamiana,* two protein bands were detected that, as mentioned earlier, may correspond to the processed and unprocessed forms of the protease

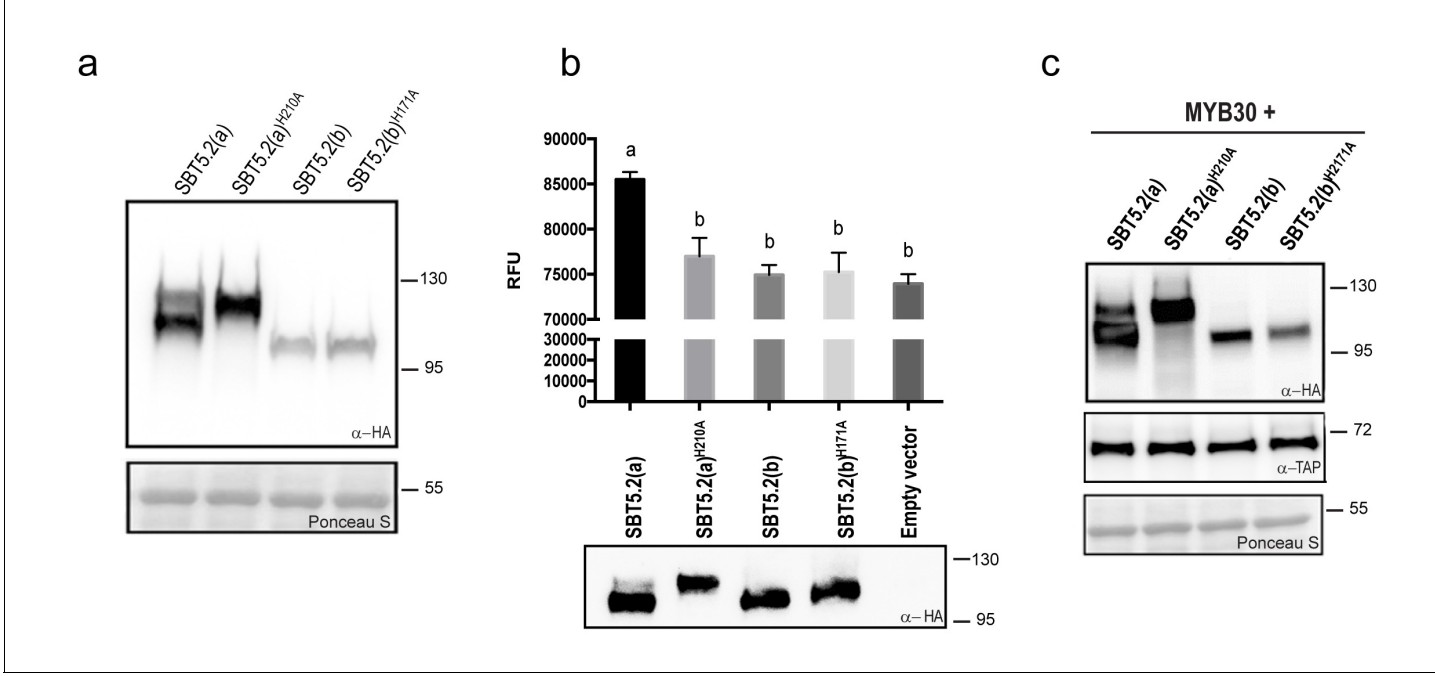

**Figure 4.** SBT5.2(a), but not SBT5.2(b), shows serine protease activity. (**a**) Western blot analysis shows expression of HA-tagged SBT5.2(a), SBT5.2(b) and their catalytic mutant versions in *N. benthamiana*, as indicated. Ponceau S staining confirms equal loading. Molecular mass markers in kilodaltons are indicated on the right. (**b**) Fluorimetric assay to detect protease activity following incubation of Arabidopsis protoplasts expressing the indicated proteins with fluorescein isothiocyanate (FITC)-conjugated casein (top). RFU: relative fluorescence units. Error bars indicate SEM. Lowercase letters indicate significant differences as determined by Bonferroni-corrected p-values (p<0.001) obtained after ANOVA and subsequent LSD post-hoc test. All proteins were detected by Western blot (bottom). (**c**) TAP-tagged MYB30 was expressed in *N. benthamiana* alone or with SBT5.2(a), SBT5.2(b) and their catalytic mutant versions, as indicated. Western blot analysis shows the expression of TAP-tagged MYB30 and HA-tagged SBT proteins. Ponceau S staining confirms equal loading. Molecular mass markers in kilodaltons are indicated on the right.

(*Figure 4a*). In contrast, only a single band was detected for SBT5.2(b), suggesting that this protein is either not processed or fully processed *in planta* (*Figure 4a*). In order to learn more about the proteolytic cleavage of SBT5.2 proteins, we engineered SBT5.2(a) and SBT5.2(b) mutant versions, in which the conserved histidine residue in the catalytic triad of both proteins was mutated to alanine, [SBT5.2(a)$^{H210A}$ and SBT5.2(b)$^{H171A}$]. Following transient expression in *N. benthamiana*, mutation of the catalytic histidine residue did not affect migration of SBT5.2(b) as compared to the wild-type protein, suggesting that this protein does not self-process *in planta* (*Figure 4a*). In contrast, in the case of SBT5.2(a)$^{H210A}$, only the slow migrating band, that very likely corresponds to the unprocessed form of the protein, was detected (*Figure 4a*). This observation suggests that SBT5.2(a) is able to auto-process *in planta* and is thus active as a protease.

The catalytic activity of HA-tagged SBT5.2(a) and SBT5.2(b) was further investigated using a fluorimetric assay. Protein extracts from Arabidopsis protoplasts expressing HA-tagged SBT5.2(a), SBT5.2(a)$^{H210A}$, SBT5.2(b) or SBT5.2(b)$^{H171A}$ were incubated with the generic protease substrate casein conjugated to fluorescein isothiocyanate (FITC). Increased fluorescence was observed for SBT5.2(a), reflecting proteolytic cleavage of the FITC-casein substrate (*Figure 4b*). In contrast, fluorescence intensity in the case of SBT5.2(a)$^{H210A}$, SBT5.2(b) and SBT5.2(b)$^{H171A}$ was indistinguishable from that of protoplasts transformed with an empty vector (*Figure 4b*), although their protein expression levels were comparable to those of SBT5.2(a) (*Figure 4b*). These results reinforce the idea that SBT5.2(a), but not SBT5.2(b), is correctly processed thus displaying protease activity.

To determine whether SBT5.2(a) or SBT5.2(b) are involved in MYB30 proteolytic processing, the *in planta* accumulation of MYB30 when expressed alone or together with the different SBT5.2 proteins was analysed. As shown in *Figure 4c*, MYB30 accumulation was consistently unaltered in the presence of SBT5.2(a) or SBT5.2(b) as compared to the expression observed in the presence of the

respective subtilase catalytic mutant versions. These results suggest that neither SBT5.2(a) nor SBT5.2(b) are able to proteolytically cleave MYB30.

## SBT5.2(b), but not SBT5.2(a), interacts with MYB30 at intracellular vesicles

MYB30 was previously localized to the nucleus of *N. benthamiana* and Arabidopsis cells (*Froidure et al., 2010*). In order to investigate MYB30 potential colocalization with SBT5.2(a) and/or SBT5.2(b), GFP-tagged MYB30 was co-expressed with RFP-tagged SBT5.2(a) or SBT5.2(b). Confocal microscopy analysis of *N. benthamiana* leaves transiently co-expressing GFP-MYB30 and SBT5.2(a)-RFP showed that these proteins do not co-localize *in planta*, as SBT5.2(a) and MYB30 retain their respective extracellular and nuclear localization when expressed together (*Figure 5a*). Surprisingly, when co-expressed with RFP-tagged SBT5.2(b), GFP-MYB30 was excluded from the nucleus and localized to the same vesicle-like structures where SBT5.2(b) was localized, suggesting a possible *in planta* interaction between the two proteins outside the nucleus (*Figure 5a*). Importantly, the unrelated MYB TF MYB123 retained its nuclear localization when co-expressed with SBT5.2(b), suggesting that SBT5.2(b)-mediated MYB30 nuclear exclusion is specific (*Figure 5a*). Moreover, SBT5.2(b)-mediated specific nuclear exclusion of MYB30 was confirmed in Arabidopsis protoplasts (*Figure 5b*).

We next sought out to confirm the interaction between MYB30 and SBT5.2 in plant cells. We focused on the study of the interaction between MYB30 and SBT5.2(b) because we were unable to detect a subcelullar co-localisation between MYB30 (nuclear) and SBT5.2(a) (secreted) (*Figure 5a,b*), which is a first requisite for the study of protein-protein interactions. The physical interaction between SBT5.2(b) and MYB30 was investigated in FRET-FLIM assays using GFP- (donor) and RFP- (acceptor) tagged MYB30 and SBT5.2(b), respectively. In order to avoid potential changes in GFP lifetime due to differences in the molecular environments of two distinct subcellular compartments [MYB30 being nuclear when expressed alone or in vesicle-like structures when co-expressed with SBT5.2(b)], the subcellular localization of GFP-tagged MYB30 when co-expressed with non-fluorescent HA-tagged, or an untagged version of SBT5.2(b), was therefore investigated. Importantly, both SBT5.2(b)-HA and untagged SBT5.2(b), which are not able to act as acceptors for GFP fluorescence, also led to MYB30 retention in intracellular vesicle-like structures (*Figure 5—figure supplement 1a, b*). A significant reduction of GFP lifetime was observed when GFP-MYB30 was co-expressed with RFP-tagged SBT5.2(b) as compared to co-expression with SBT5.2(b)-HA, both in *N. benthamiana* epidermal cells and in Arabidopsis protoplasts, thus confirming the physical interaction between the two proteins in intracellular vesicles (*Figure 5c,d*; *Table 1*). This interaction did not depend on the integrity of SBT5.2(b) catalytic triad, as shown by the reduced GFP lifetime of GFP-MYB30 when co-expressed with RFP-tagged SBT5.2(b)[H171A] (*Table 1*; *Figure 5—figure supplement 1c*).

The identification of a partial *SBT5.2* cDNA clone in yeast suggested that the MYB30-SBT5.2(b) interaction is mediated by the C-terminus of SBT5.2(b). In order to confirm this idea, a truncated SBT5.2(b) version containing the C-terminal end of the protein [SBT5.2(b)[362-730]] fused to the RFP was generated and transiently expressed in *N. benthamiana*. SBT5.2(b)[362-730]-RFP presents a nucleo-cytoplasmic localization and colocalises with MYB30 in the nucleus (*Figure 5—figure supplement 2a*). A significant reduction of the average GFP lifetime was measured in nuclei coexpressing GFP-MYB30 and SBT5.2(b)[362-730]-RFP, as compared to nuclei expressing GFP-MYB30 alone (*Figure 5—figure supplement 2d*; *Table 1*), confirming that the C-terminus of SBT5.2(b) is sufficient for the interaction with MYB30. The specificity of this observation was highlighted by the lack of interaction between GFP-MYB30 and the equivalent C-terminal domain of the closest Arabidopsis SBT5.2 homolog, SBT5.1, (SBT5.1[405-780]-RFP) (*Figure 5—figure supplement 2b,e*; *Table 1*). Moreover, no significant reduction of the average GFP lifetime was detected between nuclei expressing the unrelated TF MYB123 [whose nuclear localization was not affected in the presence of full length SBT5.2(b) (*Figure 5a,b*)], when expressed alone or together with SBT5.2(b)[362-730]-RFP, despite the nuclear co-localization of the two proteins (*Figure 5—figure supplement 2c,f*; *Table 1*).

Together, our data confirms that MYB30 specifically interacts with SBT5.2(b) at vesicle-like structures. This interaction is mediated by SBT5.2(b) C-terminus, does not require an intact SBT5.2(b) catalytic triad and results in MYB30 nuclear exclusion.

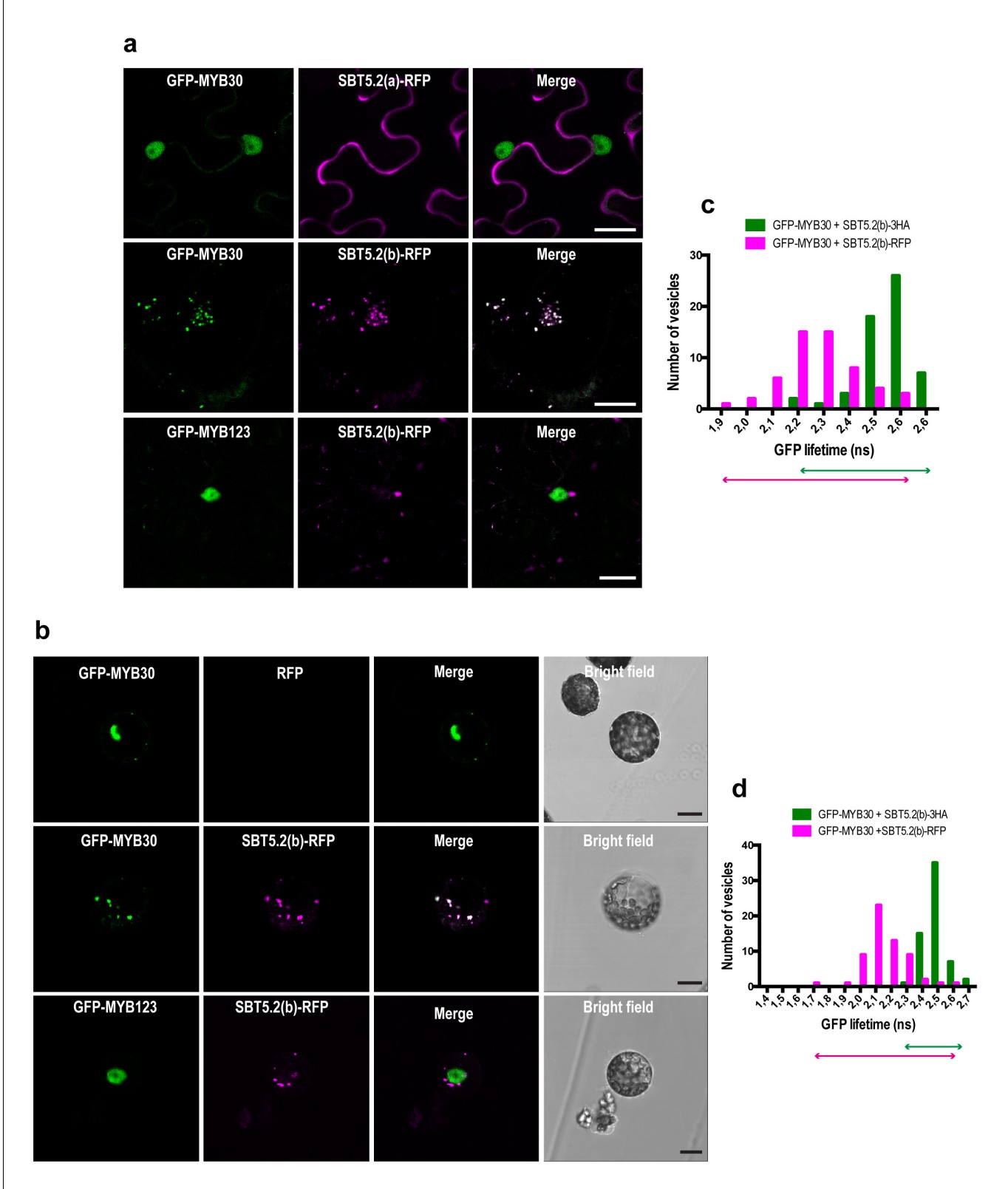

**Figure 5.** SBT5.2(b) mediates retention of MYB30 in intracellular vesicles. (a) Confocal images of epidermal *N. benthamiana* cells 36 hr after *Agrobacterium*-mediated transient expression of the indicated constructs. (b) Confocal images of Arabidopsis protoplasts 16 hr after transformation of the indicated constructs. (c,d) GFP lifetime distribution of GFP-MYB30 in *N. benthamiana* cells (c) or Arabidopsis protoplasts (d) expressing SBT5.2(b). Histograms show the number of vesicles according to GFP-MYB30 lifetime classes in the presence of SBT5.2(b)-HA (green bars) or SBT5.2(b)-RFP

*Figure 5 continued on next page*

*Figure 5 continued*

(magenta bars). The degree of overlap of GFP lifetime distribution is represented with magenta (SBT5.2(b)-RFP) and green (SBT5.2(b)-HA) arrows. Bars = 10 μm.

The following figure supplements are available for figure 5:

**Figure supplement 1.** SBT5.2(b)-mediated retention of MYB30 outside the nucleus is independent of C-terminal tagging of the subtilase and of SBT5.2 (b) catalytic triad.

**Figure supplement 2.** MYB30 interacts with SBT5.2(b) through the C-terminus of the subtilase.

## An N-terminal myristoylation site in SBT5.2(b) determines its localization to endosomes and MYB30 nuclear exclusion

We next sought to determine the nature of the intracellular vesicle-like structures where SBT5.2(b) resides. Given the mobile character and varied sizes of these vesicles, we conducted co-localization experiments with VHA-a1 and SYP61, two markers for the trans-Golgi network/early endosomes (TGN/EE) (*Dettmer et al., 2006*; *Tanaka et al., 2009*) and ARA6 and SYP21, two markers for late endosomes/multivesicular bodies (LE/MVB) (*Ueda et al., 2004*; *Uemura et al., 2004*). Colocalization with of SBT5.2(b) with these endosomal markers was clearly observed in *N. benthamiana* leaves (*Figure 6a*, *Figure 6—figure supplement 1*). In contrast, no colocalization of SBT5.2(b)-RFP with the Golgi marker GmMan-GFP (*Nelson et al., 2007*) was detected (*Figure 6a*).

The nucleocytoplasmic subcellular localization of SBT5.2(b)$^{362-730}$ suggested that the N-terminal region of SBT5.2(b) is required for its localization to endosomes (*Figure 5—figure supplement 2*). In order to further test this idea, an N-terminal deletion of SBT5.2(b) was generated [SBT5.2(b)$^{162-730}$]. RFP-tagged SBT5.2(b)$^{162-730}$ indeed localized to the cytoplasm (*Figure 6b*). Furthermore, an N-terminally tagged RFP fusion of SBT5.2(b) also presented a cytoplasmic localization (*Figure 6b*), suggesting that a free SBT5.2(b)N-terminus is necessary for SBT5.2(b) endosomal targeting. Close

**Table 1.** FRET-FLIM analysis shows that MYB30 physically interacts with SBT5.2(b) at intracellular vesicles.

| Donor | Acceptor | Lifetime* | SD[†] | N[‡] | E[§] | p-value[#] |
|---|---|---|---|---|---|---|
| *N. benthamiana* | | | | | | |
| GFP-MYB30 | SBT5.2(b)-HA | 2.552 | 0.013 | 57 | - | |
| GFP-MYB30 | SBT5.2(b)-RFP | 2.274 | 0.019 | 54 | 10.86 | $5.8 \times 10^{-21}$ |
| GFP-MYB30 | - | 2.669 | 0.009 | 82 | | |
| GFP-MYB30 | SBT5.2(b)$^{362-730}$-RFP | 2.271 | 0.016 | 51 | 14.91 | $3.8 \times 10^{-49}$ |
| GFP-MYB30 | SBT5.1(b)$^{405-780}$-RFP | 2.592 | 0.018 | 44 | 2.86 | $4.2 \times 10^{-05}$ |
| GFP-MYB123 | - | 2.570 | 0.013 | 59 | | |
| GFP-MYB123 | SBT5.2(b)$^{362-730}$-RFP | 2.544 | 0.013 | 58 | 1.00 | 0.17 |
| *A. thaliana* | | | | | | |
| GFP-MYB30 | SBT5.2(b)-HA | 2.491 | 0.009 | 60 | - | |
| GFP-MYB30 | SBT5.2(b)-RFP | 2.151 | 0.018 | 60 | 13.65 | $1.2 \times 10^{-33}$ |
| GFP-MYB30 | SBT5.2(b)$^{H171A}$-HA | 2.473 | 0.012 | 60 | - | |
| GFP-MYB30 | SBT5.2(b)$^{H171A}$-RFP | 2.115 | 0.023 | 60 | 14.49 | $7.8 \times 10^{-26}$ |

* Mean lifetime in nanoseconds

[†] Standard deviation

[‡] Total number of measured vesicles

[§] Percentage of FRET efficiency (E = 1 - τDA/τD) calculated by comparing the lifetime of the donor in the presence of the acceptor (τDA) with its lifetime in the absence of the acceptor (τD).

[#] p value of the difference between the donor lifetimes in the presence and in the absence of the acceptor (*t*-test)

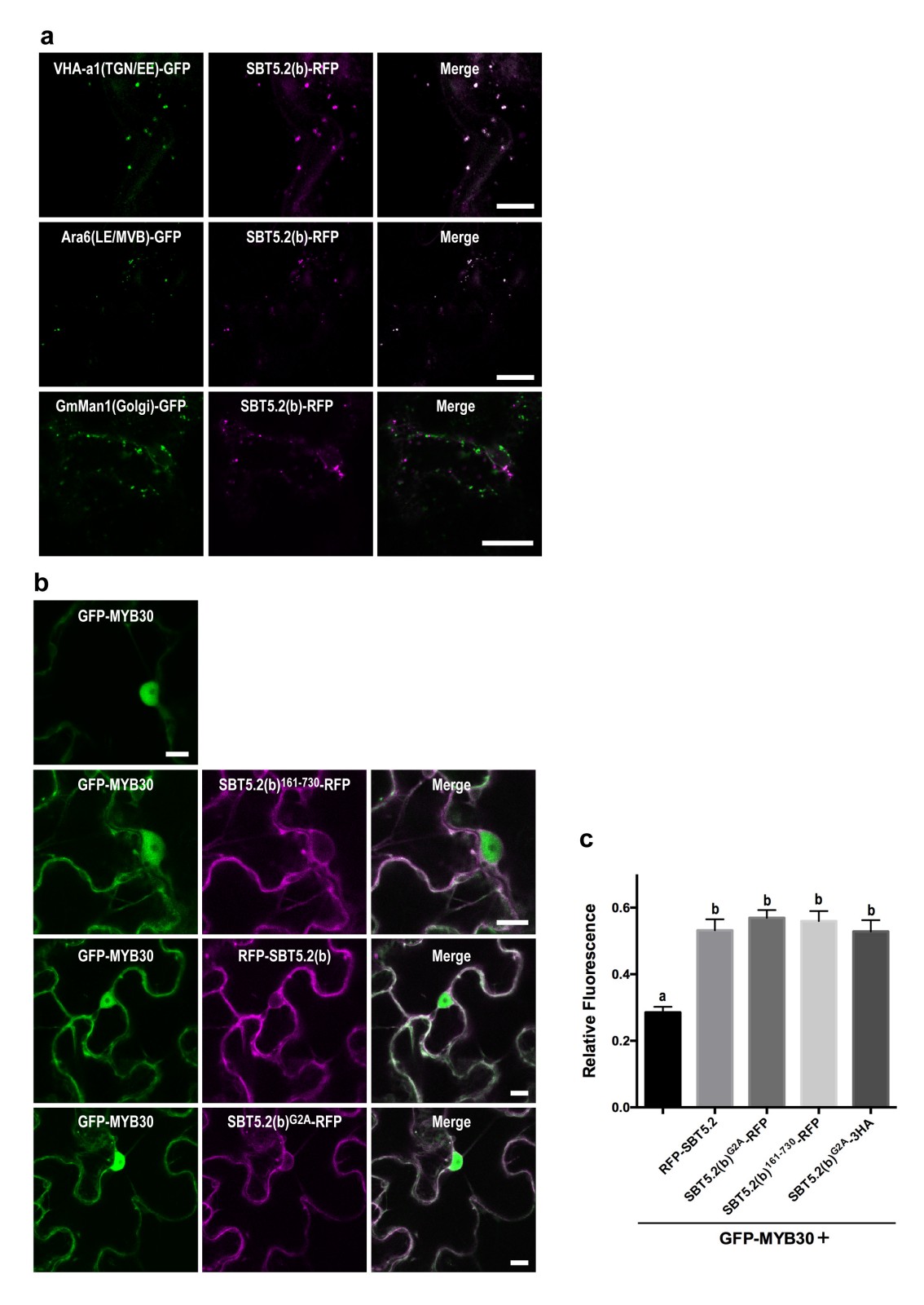

**Figure 6.** An N-terminal myristoylation site in SBT5.2(b) is required for its localization to endosomes and MYB30 nuclear exclusion. (a,b) Confocal images of epidermal *N. benthamiana* cells 36 hr after *Agrobacterium*-mediated transient expression of the indicated constructs. Bars = 10 μm. (a) SBT5.2(b) localizes to endosomal vesicles. TGN/EE: trans-Golgi network/early endosomes; LE/MVB: late endosomes/multivesicular bodies. (b) A free N-terminal myristoylation site in SBT5.2(b) mediates its localization to endosomes and retention of MYB30 in endosomal vesicles. (c) Relative

*Figure 6 continued on next page*

*Figure 6 continued*

fluorescence values of cytoplasmic MYB30, expressed alone or with the indicated SBT5.2(b) versions, represented as ratios between cytoplasmic and nuclear fluorescence values in individual cells. Mean and SEM values were calculated from two independent experiments in which fifteen fluorescence measurements were taken per experiment and construct combination. Lowercase letters indicate statistically significant differences as determined by Bonferroni-corrected p-values (p<0.001) obtained after ANOVA and subsequent LSD post-hoc test.

The following figure supplement is available for figure 6:

**Figure supplement 1.** SBT5.2(b) localizes to endosomal vesicles.

inspection of the SBT5.2(b)N-terminal region uncovered the presence of a putative myristoylation site (MGSASSA; *Figure 2—figure supplement 1c*). A SBT5.2(b) version in which the putatively myristoylated Gly2 residue was mutated to Ala [SBT5.2(b)$^{G2A}$] also localized to the cytoplasm (*Figure 6b*), confirming the importance of N-myristoylation for SBT5.2(b) endosomal targeting. Finally, co-expression of SBT5.2(b)$^{162-730}$-RFP, RFP-SBT5.2(b) or SBT5.2(b)$^{G2A}$ RFP with GFP-MYB30 did not affect MYB30 nuclear targeting (*Figure 6b*). Interestingly, MYB30 accumulation in the cytoplasm was enhanced in the presence of these SBT5.2(b) versions, consistent with the presence in the three proteins of an intact SBT5.2 C-terminal domain that mediates the interaction with MYB30 (*Figure 6c*). An HA-tagged SBT5.2(b)$^{G2A}$ version induced the same effect confirming that the enhanced detection of GFP-MYB30 in the cytoplasm is not an artefact due the presence of a second fluorophore. Together, our data strongly suggest that a free N-terminal myristoylated residue is responsible of SBT5.2(b) targeting to the endosomes and of MYB30 retention in endosomal vesicles.

In order to further characterize the simultaneous localization of SBT5.2(b) to both early and late endosomes, HA-tagged versions of either SBT5.2(a), SBT5.2(b) or SBT5.2(b)$^{G2A}$ were co-expressed with both GFP-tagged VHA-a1 and RFP-tagged ARA6. When expressed with SBT5.2(a), both subcellular markers conserved their distinct subcellular localization in TGN/EE and LE/MVB, respectively (*Figure 7*). In contrast, co-expression with SBT5.2(b) led to co-localization of VHA-a1 and ARA6 in the same endosomal compartment, strongly suggesting that SBT5.2(b) is able to interfere with endosomal trafficking. Myristoylation of SBT5.2(b) appears to be essential to this effect, since expression of SBT5.2(b)$^{G2A}$ did not affect the discrete endosomal populations tagged by VHA-a1 or ARA6 (*Figure 7*).

## SBT5.2(b) attenuates MYB30-mediated responses to bacterial infection

To investigate the function of SBT5.2 in the plant response to bacterial inoculation, we used Arabidopsis *sbt5.2* null mutants, *sbt5.2–1* (SALK_012113) and *sbt5.2–2* (SALK_132812C), both containing a T-DNA insertion in the last exon of *SBT5.2*. Despite the severe reduction of *SBT5.2* expression in the mutant lines (*Figure 8—figure supplement 1*), no obvious macroscopic phenotype was observed in these plants. The phenotype of these lines in response to bacterial inoculation was next analysed. Similar to MYB30-overexpressing (MYB30$_{OE}$), *sbt5.2* mutant plants showed stronger HR cell death symptoms after inoculation with *Pst AvrRpm1* as compared to Col-0 wild-type plants (*Figure 8a*). This phenotype was quantified by ion leakage measurements in leaf disk assays. Conductivity values measured in *sbt5.2* and MYB30$_{OE}$ plants were significantly higher than those displayed by Col-0 wild type plants after bacterial inoculation (*Figure 8b*). In agreement with faster HR development, *sbt5.2* plants showed increased resistance in response to inoculation with *Pst AvrRpm1*, as compared to wild-type plants (*Figure 8c*), confirming the role of SBT5.2 as a negative regulator of plant defence.

Importantly, *sbt5.2* mutant plants displayed higher expression of MYB30 VLCFA-related target genes *FDH and CER2* (*Raffaele et al., 2008*) as compared to Col-0 wild-type plants 1 hr after inoculation (*Figure 9a*). Moreover, this phenotype was abolished in the *myb30* mutant background (*sbt5.2 myb30*; *Figure 9a*) and correlated with loss of increased HR in the *sbt5.2 myb30* double mutant (*Figure 9b*). Together, these data confirm that SBT5.2 negatively regulates Arabidopsis defence through repression of MYB30 transcriptional activity.

In *sbt5.2* mutant plants, expression of both *SBT5.2(a)* and *SBT5.2(b)* is affected. To obtain additional proof of the negative role specifically played by SBT5.2(b) on MYB30-mediated responses, *sbt5.2* mutant plants were transformed with an HA-tagged version of either SBT5.2(a) or SBT5.2(b)

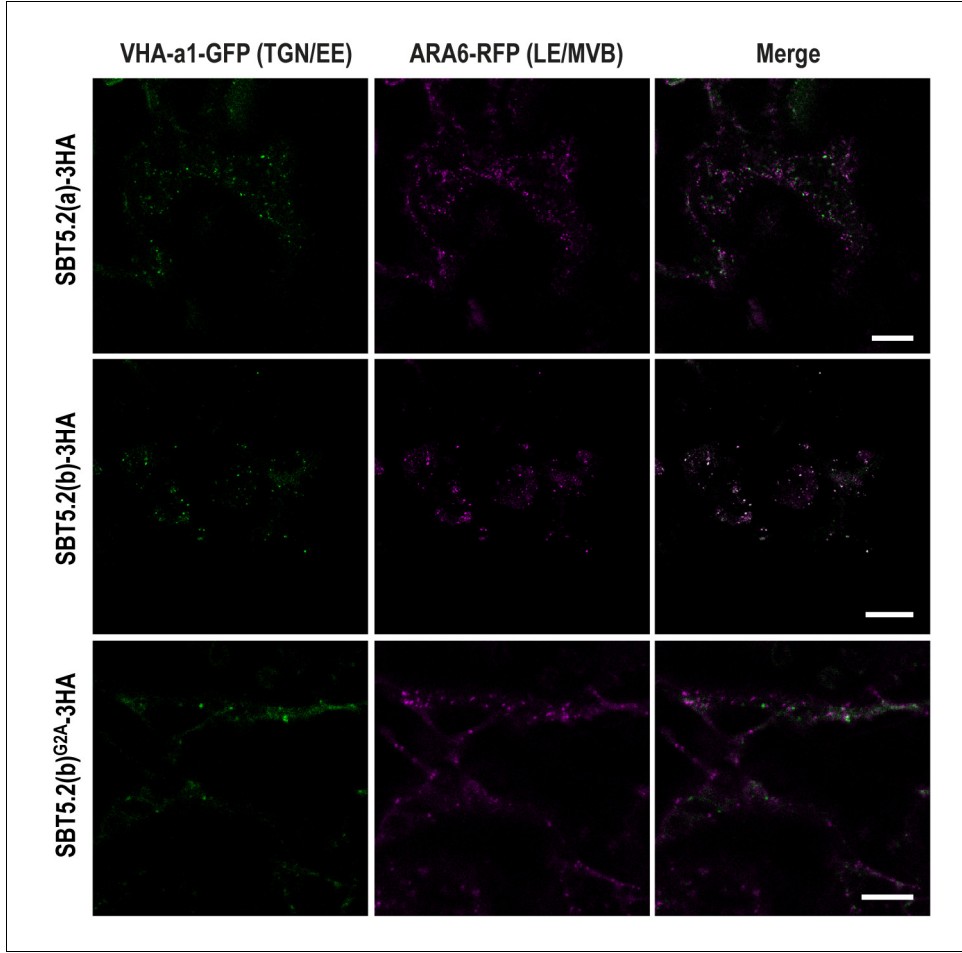

**Figure 7.** SBT5.2(b) leads to the formation of hybrid endosomal compartments in a myristoylation-dependent manner. Confocal images of epidermal *N. benthamiana* cells 36 hr after *Agrobacterium*-mediated transient expression of the indicated constructs. TGN/EE: trans-Golgi network/early endosomes; LE/MVB: late endosomes/ multivesicular bodies. Bars = 10 µm.

under the control of the 35S promoter. Expression of *SBT5.2* gene and protein was monitored by qRT-PCR and Western Blot analysis in two independent homozygous T4 lines for each construct (***Figure 8—figure supplement 1b,c***). Importantly, the increased HR phenotype displayed by the *sbt5.2* mutant was specifically suppressed in *sbt5.2* plants overexpressing SBT5.2(b), but not SBT5.2(a) (***Figure 9c***).

In order to obtain *in vivo* confirmation of the negative control of defence-related cell death exerted by SBT5.2(b), the above described lines were inoculated with a low dose of HR-inducing *Pst AvrRpm1*. Enhanced accumulation of phenolic compounds, characteristic of HR cell death, was detected in $MYB30_{OE}$ ans *sbt5.2* mutant lines as compared to wild-type Col-0 and this phenotype was suppressed both in *sbt5.2 myb30* double mutants and by overexpressing SBT5.2(b), but not SBT5.2(a), in the *sbt5.2* mutant background (***Figure 9d,e***). Overall our data demonstrate that, in agreement with SBT5.2(b)-mediated retention of MYB30 in endosomes, negative regulation of MYB30-mediated defence-related cell death is specifically controlled by SBT5.2(b).

## Discussion

AS is a fundamental process that allows generating a large number of mRNA and protein isoforms from a genome of limited size (***Graveley, 2005***). Moreover, AS plays crucial functions in eukaryotic cells, as it determines the binding properties, intracellular localisation, enzymatic activity, protein

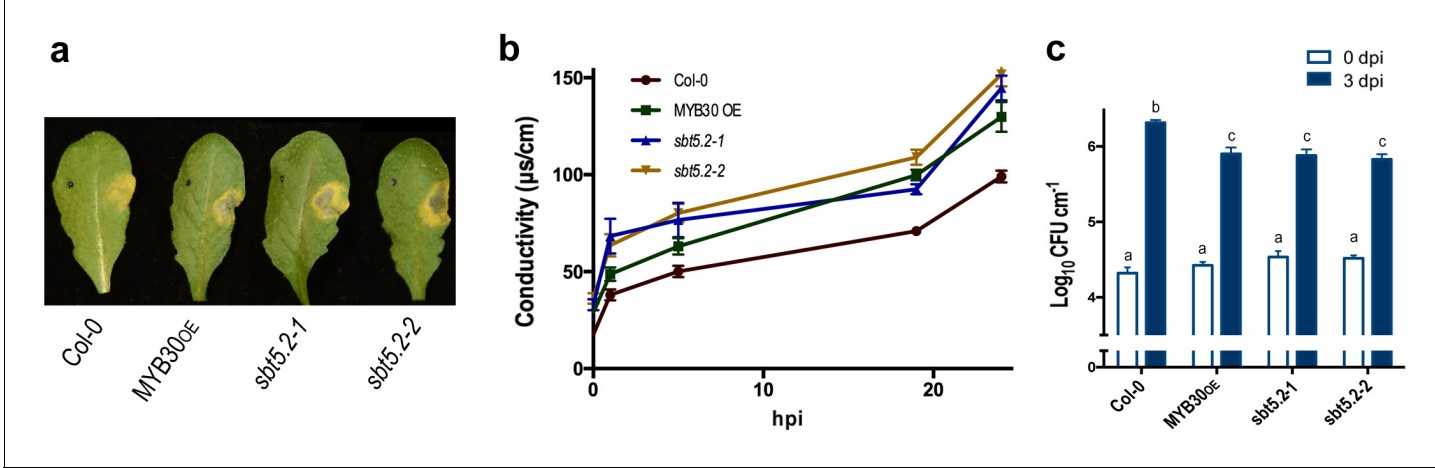

**Figure 8.** SBT5.2(b) is a negative regulator of resistance and HR responses in Arabidopsis in response to bacterial inoculation. (**a**) Symptoms developed by the indicated Arabidopsis lines 60 hpi with *Pst AvrRpm1* (2 × 10⁶ cfu/ml). The pictures are representative of three independent experiments in which 4 plants of each line were infiltrated. (**b**) Quantification of cell death by measuring electrolyte leakage of the indicated Arabidopsis lines in a time course of 24 hr. Plants were inoculated with *Pst AvrRpm1* (5 × 10⁶ cfu/ml). Mean and SEM values were calculated from four independent experiments in which three plants were inoculated (four leaves/plant). (**c**) Growth of *Pst AvrRpm1* in the indicated Arabidopsis lines. Bacterial growth 0 (white bars) and three days (blue bars) was measured after inoculation (5 × 10⁵ cfu/ml). Mean bacterial densities were calculated from 6 independent experiments with 6 individual plants (4 leaves/plant). Statistical differences using multiple factor analysis of variance (ANOVA) (p<0.001) are indicated by letters.

The following figure supplement is available for figure 8:

**Figure supplement 1.** Characterization of *sbt5.2* mutant lines before and after transformation with SBT5.2(a) and SBT5.2(b) overexpresing constructs.

stability and posttranslational modifications of a large number of proteins (*Stamm et al., 2005*). In humans, the importance of AS is clearly highlighted by the fact that about 15% of genetic hereditary diseases are caused by mutations that affect splicing (*Kornblihtt et al., 2013*). In plants, AS plays fundamental roles in regulating plant growth, development, and responses to environmental signals (*Staiger and Brown, 2013*). A genome-wide transcriptomic analysis in Arabidopsis plants inoculated with bacteria uncovered a surprisingly large number of AS events (*Howard et al., 2013*). Although we are still far from understanding the functional implications of this transcriptome complexity, the function of numerous plant resistance genes, encoding immune receptors, appears to be regulated by AS (*Yang et al., 2014*; *Gassmann, 2008*). To our knowledge, our work represents the first described example of AS affecting the function of a protein of the subtilase family. AS of *SBT5.2* results in the production of the atypical subtilase SBT5.2(b), uncovering a novel mode of regulation of defence reactions and contributing to further our understanding of the varied roles of AS in the control of plant immunity.

Following bacterial inoculation, the respective induction and repression of *SBT5.2(b)* and *SBT5.2(a)* expression suggests a functional role for SBT5.2(b) during defence regulation. This idea is reinforced by the observed co-regulation of *SBT5.2(b)* and *MYB30* expression after bacterial treatment. Despite their pervasiveness, our current understanding of the functions of plants subtilases is still limited. Different studies suggest a role in both general protein turnover and regulation of plant development or responses to environmental cues (*Schaller et al., 2012*; *Figueiredo et al., 2014*). The first example of a plant subtilase potentially acting during plant-pathogen interactions was reported in tomato, where the expression of the subtilases *P69B* and *P69C* was induced by pathogen infection and treatment with salycilic acid (*Jordá et al., 1999*; *Tornero et al., 1997*). More recently, the Arabidopsis subtilase SBT3.3 was found to be involved in the regulation of immune signalling through chromatin remodelling of defence-related genes associated with the activation of immune priming (*Ramírez et al., 2013*).

Plant subtilases are usually synthesized in the form of preproprotein precursors, translocated *via* an N-terminal ER-targeting SP into the endomembrane system (*Schaller et al., 2012*). In addition, using mass spectrometry (*Cedzich et al., 2009*) and structural analyses (*Murayama et al., 2012*),

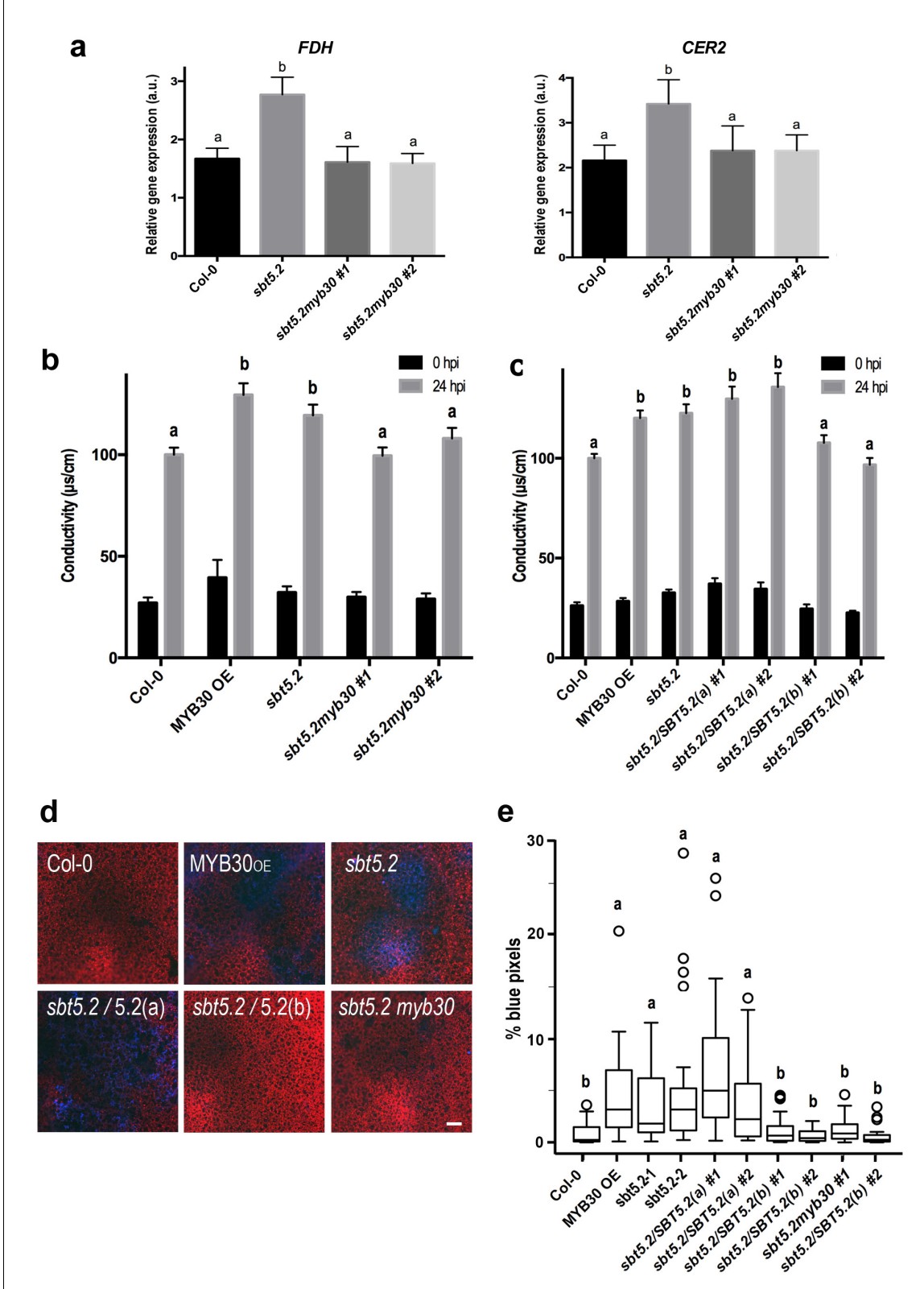

**Figure 9.** SBT5.2(b) attenuates MYB30-dependent transcriptional activation of VLCFA-related genes and hypersensitive cell death. (a) Expression analysis of the MYB30 target genes *FDH* and *CER2* in the indicated Arabidopsis lines 1 hr after inoculation with *Pst AvrRpm1* ($5 \times 10^7$ cfu/ml). Expression values of the individual genes were normalized using *SAND* family as internal standard. Mean and SEM values were calculated from 3 independent experiments (4 replicates/experiment). Statistical significance according to a Student's *t*-test (p<0.05) is indicated by letters. (**b,c**)
*Figure 9 continued on next page*

*Figure 9 continued*

Quantification of cell death by measuring electrolyte leakage of the indicated Arabidopsis lines before (black bars) and 24 hr after (gray bars) inoculation with *Pst AvrRpm1* (5 × 10$^6$ cfu/ml). Mean and SEM values were calculated from four independent experiments (three plants/experiment and four leaves/plant) and related to the value displayed by wild-type Col-0 plants, which is set at 100%. Statistical differences using multiple factor analysis of variance (ANOVA) (p<0.01) are indicated by letters. (d) Representative pictures of accumulation of phenolic compounds (blue coloration indicative of cell death) in the indicated Arabidopsis lines detected by epifluorescence 24 hr after inoculation with *Pst AvrRpm1* (2 × 10$^5$ cfu/ml). Bar = 100 µm. (e) Blue pixels in the indicated lines were quantified using Image-Pro Plus and are shown as the percentage of the total number of blue pixels in each image. Boxplots are as follows: box limits, values between first and third quartiles; middle bar, median. Whiskers cover 1.5 times the interquartile distance and circles represent extreme values. Lowercase letters indicate significant differences with respect to the *sbt5.5–2* line as determined by Bonferroni-corrected p-values (p<0.01) obtained after ANOVA and subsequent LSD post-hoc test.

plant subtilases were shown to be glycosylated in the secretory pathway and to accumulate extracellularly (*Schaller et al., 2012*). In agreement with the preproprotein structure, the maturation of the active enzyme from its inactive precursor requires at least two processing steps. After cleavage of the SP, subtilases are ultimately activated by cleavage of the PD producing the mature active enzyme (*Taylor et al., 1997*). Processing of the PD, an auto-inhibitor domain of plant subtilases (*Nakagawa et al., 2010*; *Meyer et al., 2016*), is an intramolecular autocatalytic reaction that occurs late in the ER or in the early Golgi (*Cedzich et al., 2009*). Here, we show that AS of the *SBT5.2* gene has important implications for the subcellular localization and activity of the resulting isoforms, despite the nearly total conservation between the two protein sequences. Consistent with harbouring all canonical features of a subtilase, we confirmed that SBT5.2(a) enters the secretory pathway, where it is glycosylated, and is secreted to the extracellular space as shown before (*Engineer et al., 2014*; *Kaschani et al., 2012*). SBT5.2 was previously described as a negative regulator of stomatal density under high $CO_2$ conditions through cleavage of the extracellular pro-peptide ligand EPF2 in the apoplast (*Engineer et al., 2014*). In contrast, in agreement with its lacking a SP and the first five amino acids of the PD, SBT5.2(b) does not enter the secretory pathway, its PD is thus not cleaved and may fold onto SBT5.2(b) catalytic domain inhibiting its protease activity. Indeed, despite the presence of PGSs and catalytic residues, we were unable to detect SBT5.2(b) glycosylation or protease activity. We also showed that MYB30 accumulation *in planta* is not affected by SBT5.2(a) or SB5.2(b), indicating that these proteins do not proteolytically cleave the TF. In the case of catalytically active SBT5.2(a), this can be explained by the lack of co-localization of both proteins. Despite their subcellular co-localization and physical interaction, lack of modification of MYB30 accumulation in the presence of SBT5.2(b) is consistent with SBT5.2(b) being inactive as a protease, as underlined by the finding that both wild-type SBT5.2(b) and catalytic mutant SBT5.2(b)[H171A] are able to interact with and retain MYB30 at endosomal vesicles. These results suggest an alternative mode of action of SBT5.2(b) on MYB30 activity, likely related to SBT5.2(b)-mediated MYB30 nuclear exclusion. Examples of proteases playing proteolysis-independent functions have been described in mammals. The proprotein convertase subtilisin/kexin type 9 (PCSK9) plays a major regulatory role in cholesterol homeostasis but does not require its catalytic activity to induce degradation of low-density lipoproteins (LDLs) (*McNutt et al., 2007*). In addition, γ-secretase, an aspartyl protease that performs the final proteolytic cleavage step in the processing cascade of the β-amyloid precursor protein (βAPP) (*Vassar et al., 1999*), displays proteolysis-independent functions during the regulation of βAPP maturation and trafficking (*Wrigley et al., 2005*).

SBT5.2(b) localises to both early and late endosomal compartments and this requires N-terminal myristoylation of the protein. Since co-expression of SBT5.2(b), but not SBT5.2(a), with both early and late endosome markers leads to colocalisation of both markers, it is tempting to speculate that SBT5.2(b) is able to interfere with endosomal trafficking leading to the formation of hybrid endosomes. In mammals, expression of constitutively active Rab5 blocks the conversion of early to late endosomes, giving rise to hybrid endosomal compartments and deregulation of autophagy (*Rink et al., 2005*). Similarly, overexpression of Arabidopsis sorting nexin SNX2b leads to the formation of large SNX2-containing endosome aggregations, which inhibits vesicle trafficking (*Phan et al., 2008*).

SBT5.2(b)-mediated regulation of MYB30 activity supports the emerging idea that endosomal trafficking pathways are not only central regulators of plasma membrane protein homeostasis but

also control multiple signalling pathways, including those involved in plant disease resistance (*Reyes et al., 2011*). Enhanced susceptibility of plants impaired in regulators of endosome trafficking emphasizes the importance of endocytic processes in establishing defence (*Lu et al., 2012*). In addition, plasma membrane receptors that sense apoplastic microbes are immune-related cargos of plant trafficking pathways (*Beck et al., 2012*; *Ben Khaled et al., 2015*), and a number of intracellular immune receptors have been shown to constitutively localize to endomembranes (*Takemoto et al., 2012*; *Engelhardt et al., 2012*), further underlining the prominent role of endomembrane systems in determining plant resistance. Although the role of endosomes in implementing defence signalling is not yet well understood, they have been proposed as central subcellular sites for orchestration of the HR. Indeed, relocalisation of the potato resistance protein R3a from the cytoplasm to endosomal compartments is required to trigger the HR (*Engelhardt et al., 2012*). The importance of these compartments during immunity is further underlined by the fact that both plant and animal pathogens produce inhibitory effector proteins that specifically target endomembrane trafficking. For example, the Arabidopsis ADP ribosylation factor (ARF)-Guanine exchange factor (GEF) MIN7 is targeted and degraded by the bacterial effector HopM1 altering secretory trafficking (*Nomura et al., 2006*, *2011*). In mammalian cells, the effector PipB2 from the bacterial pathogen *Salmonella enterica* causes a specific redistribution of late endosomes/lysosomes (LE/Lys) compartments to the cell periphery promoting formation of *Salmonella*-induced filaments (Sifs) (*Knodler and Steele-Mortimer, 2005*).

Our finding that SBT5.2(b) mediates MYB30 nuclear exclusion resulting in attenuation of MYB30 gene expression and HR, is in agreement with several reports describing controlled nuclear localisation as an efficient mechanism to regulate TF activity in eukaryotic cells. Subcellular compartmentalization of the E2F TF family, either in the nucleus or in the cytoplasm, is used to control cell cycle in differentiated skeletal cells (*Gill and Hamel, 2000*). p53, a tightly regulated animal TF that acts as a hub for numerous signalling pathways including apoptosis, shifts from the nucleus to the cytoplasm in the presence of HDM2, where it has important roles that are independent of its transcriptional activity (*Boyd et al., 2000*; *Green and Kroemer, 2009*). Similarly, alternative functions for MYB30 in endosomal vesicles, other than HR regulation, cannot be excluded at this stage. In addition, nuclear import of 'dormant' TFs plays important roles in the regulation of gene expression. Upon exposure to environmental stresses, several membrane-bound TFs have been shown to be proteolytically activated by either ubiquitin-mediated proteasome activities or by specific membrane-bound proteases (*Hoppe et al., 2001*), which may facilitate triggering quick transcriptional responses to ensure plant survival under stressful conditions (*Kim et al., 2006*; *Seo et al., 2010*). Notably, subtilases are known to be involved in the release of 'dormant' membrane-bound TFs. Arabidopsis SBT6.1 is able to cleave the ER-located TF bZIP17 that, once released, moves to the nucleus to activate transcription in response to salt stress (*Liu et al., 2007*). Finally, nuclear exclusion by localization to small vesicle-like structures has been reported as a negative regulatory mechanism of TF activity. Indeed, the small interference protein MIF1, promotes nuclear exclusion of the TF ZHD5 that regulates flower architecture and leaf development. As a result, ZHD5 is relocalised into cytoplasmic vesicle-like structures, which interferes with is transcriptional activity (*Hong et al., 2011*).

During the last few years, different regulatory mechanisms of MYB30-mediated HR have been uncovered (*Raffaele and Rivas, 2013*), including spatio-temporal control of MYB30 activity through the action of the secreted phospholipase AtsPLA$_2$-$\alpha$ (that specifically relocalises to the nucleus in the presence of MYB30 [*Froidure et al., 2010*]) and the RING-type E3 ligase MIEL1 (that ubiquitinates MYB30 and leads to its proteasomal degradation [*Marino et al., 2013*]). SBT5.2(b)-mediated nuclear exclusion of MYB30 represents an additional regulatory mode of the activity of this TF, underlining the complexity of the regulatory modes of defence-related plant cell death responses. This intricate regulation of MYB30 is reminiscent of the tight control exerted on animal TFs such as p53 that is also multi-regulated through varied modes including protein-protein interactions, ubiquitination and, notably, nuclear exclusion. This sophisticated fine-tuning, which provides an efficient means to regulate fundamental cellular processes, appears as a general feature underlying the transcriptional control in eukaryotic cells.

# Materials and methods

## Cloning procedures

Plasmids used in this study were constructed by Gateway technology (GW; Invitrogen, Waltham, MA, USA) following the instructions of the manufacturer. PCR products flanked by the *att*B sites were recombined into the pDONR207 vector (Invitrogen) *via* a BP reaction to create the corresponding entry clones with *att*L sites. Inserts cloned into the entry clones were subsequently recombined into the destination vectors via an LR reaction to create the expression constructs.

A fusion of fluorescent proteins to MYB30, SBT5.2(b), SBT5.2(b)$^{G2A}$, VHA-a1, SYP61, ARA6 and SYP21 was accomplished using a multisite GATEWAY cloning strategy (Invitrogen) described previously (*Serrano et al., 2014*). Briefly, the full-length open reading frames of the indicated proteins were amplified from a plasmid template and cloned into the donor vector pBSDONR P1-P4 or the pBSDONR P4r-P2 by BP reaction (ampicillin-resistant vectors derived from pDONR221 from Invitrogen) (*Gu and Innes, 2011*). eGFP (*Cormack et al., 1996*) and RFP (*Campbell et al., 2002*) were cloned into pBSDONR P1-P4 for N-terminal fusion and into pBSDONR P4r-P2 for C-terminal fusion. To fuse SBT5.2(a), SBT5.2(b), SBT5.2(b)$^{162-730}$,SBT5.2(b)$^{G2A}$, VHA-a1, SYP61, ARA6 and SYP21 with the epitope tags, the P1-P4 clones were mixed with corresponding P4r-P2 and the desired destination vectors and recombined using Gateway LR Clonase II (Invitrogen). All the above pBSDONR constructs were recombined with the destination vector pEarleyGate100 (*Earley et al., 2006*) with the exception of SBT5.2(a) and SBT5.2(b), for which the corresponding pBSDONR constructs were recombined with the steroid-inducible destination vector pBAV154 (*Vinatzer et al., 2006*).

For yeast assays, the GAL4-BD-MYB30ΔAD fusion was previously described (*Froidure et al., 2010*). AD-SBT5.2(b)$^{628-730}$ construct was generated from recombination of the corresponding entry constructs with the pGAD-AD-GW vector (*Froidure et al., 2010*).

Point mutations were generated using the QuickChange mutagenesis kit (Stratagene, Santa Clara, CA, USA) using the pENTR-SBT5.2 as a template and following the manufacturer's instructions. Primers used for mutagenesis are shown in *Supplementary file 1*.

## Yeast assays

The yeast two-hybrid screen and methods used for identification of SBT5.2 were previously described (*Froidure et al., 2010*). Briefly, an *Arabidopsis thaliana* Gal4 yeast two-hybrid cDNA prey library (MatchMaker; Clontech) was generated from mRNA isolated from leaves of four-week-old plants (Ws-4 ecotype) syringe-infiltrated with the *Xanthomonas campestris* pv. *campestris* 147 strain. An MYB30 version deleted from its C-terminal activation domain (amino acids 1 to 234) was used as bait for screening $2 \times 10^6$ independent transformants exhibiting His auxotrophy on selective plates.

## 5' RACE assays

5' ends of *SBT5.2* mRNA were determined using the GeneRacerTM RACE Ready kit (Invitrogen, France) according to manufacturers' instructions, using RNA from Col-0 leaves and gene specific primers indicated in *Supplementary file 1*. PCR products were cloned in pGEM-T Easy vector (Promega Corporation, Fitchburg, WI, USA) and sequenced.

## Plant and bacterial materials

Arabidopsis lines used in this study were in the Columbia background and grown in Jiffy pots under controlled conditions in a growth chamber at 21°C, with a 9-hr light period and a light intensity of 190 µmol.m$^{-2}$.s$^{-1}$. The MYB30*ko* line (SALK_122884) was reported before (*Marino et al., 2013*).

For transient expression of proteins in *N. benthamiana*, overnight bacterial cultures of *Agrobacterium tumefaciens* strain C58C1 or GV3101 carrying the vector of interest were harvested by centrifugation. Cells were resuspended in induction buffer (10 mM MgCl$_2$, 10 mM MES, pH 5.6, and 150 µM acetosyringone) to an OD$_{600}$ of 0.5. After 2 hr at 22°C, cells were infiltrated into leaves of four-week-old *N. benthamiana* plants. Two days after *A. tumefaciens* infiltration, leaf discs used for experiments were harvested and processed, or frozen immediately in liquid nitrogen and stored at –80°C. For tunicamycin treatment, 24 hr after *Agrobacterium*-mediated transient expression, leaf discs of *N. benthamiana* leaves were incubated in a solution of 10 µM tunicamycin for 20 hr at room temperature and later frozen in liquid nitrogen before processing.

When testing the effect of SBT5.2 proteins on MYB30 accumulation, to minimize differences in protein expression, which are inherent to transient assays, MYB30 was co-expressed with SBT5.2(a), or SBT5.2(b), and the respective catalytic mutant side by side in the same *N. benthamiana* leaf. To avoid *ex planta* protein degradation, the leaves were pre-treated with the protease inhibitor PMSF (1 mM) 30 min before harvesting the tissue for protein extraction.

Arabidopsis four-week-old plants were kept at high humidity 12 hr before inoculation and injected with a bacterial suspension of *Pst* A*vrRpm1* at the indicated bacterial densities using a blunt syringe on the abaxial side of the leaves. For determination of *in planta* bacterial growth, the leaf samples were harvested 0 and three days after inoculation and ground on sterile water. A predetermined dilution for each sample was plated on King's B medium and incubated at 28°C for two days. The data were submitted to a statistical analysis using Statgraphics Centurion XV.II Professional Software (Statpoint Technologies Inc., Warrenton, VA, USA). Normality of residues was verified by the Kolmogorov-Smirmov test. The effect of the genotype was tested by Multiple Factor ANOVA.

## Fluorescence microscopy, FRET-FLIM and data analysis

GFP and RFP fluorescence was analyzed with a confocal laser scanning microscope (TCS SP2-AOBS; Leica) using a x63 water immersion objective lens (numerical aperture 1.20; PL APO). GFP fluorescence was excited with the 488 nm ray line of the argon laser and recorded in one of the confocal channels in the 505 to 530 nm emission range. RFP fluorescence was excited with the 561 nm line ray of the He-Ne laser and detected in the range between 595 and 620 nm. Images were acquired in the sequential mode using Leica LCS software (version 2.61).

In order to quantify the fluorescence of MYB30 in the nucleus and cytoplasm, fluorescence intensity was measured using Leica LCS software (version 2.61). Region of interest (ROIs) were defined for each photograph and the mean value was taken as a fluorescence measure. Five fluorescence measures were obtained from 6 photographs taken from two independent experiments (n = 30).

The fluorescence lifetime of the donor was experimentally measured in the presence and absence of the acceptor. The FRET efficiency (E) was calculated by comparing the lifetime of the donor in the presence ($t_{DA}$) or absence ($t_D$) of the acceptor: $E = 1-(t_{DA})/(t_D)$. Statistical comparisons between control (donor) and assay (donor + acceptor) lifetime values were performed by Student *t* test. FRET-FLIM measurements were performed using a FLIM system coupled to a streak camera (*Krishnan et al., 2003*). The light source (l = 850 nm) was a pulsed pulsed femtosecond IR laser (Spectra-Physics, USA). All images were acquired with a 60x oil immersion lens (Plan Apo 1.4 numerical aperture, IR) mounted on an inverted microscope (Eclipse TE2000E, Nikon, Japan) coupled to the FLIM system. The fluorescence emission was directed back out into the detection unit through a band pass filter. The detector was composed of a streak camera (Streakscope C4334, Hamamatsu Photonics, Japan) coupled with a fast and high sensitivity CCD camera (model C8800-53C, Hamamatsu). For each region of interest (vesicle and nucleus), average fluorescence decay profiles were plotted and lifetimes were estimated by fitting data with exponential function using a non-linear least-squares estimation procedure.

## Deglycosylation experiments

Proteins were extracted in 50 mM Tris-HCl, pH 7.5, 150 mM NaCl, 10% [v/v] glycerol, 1 mM PMSF, and 1% plant protease inhibitor cocktail (Sigma-Aldrich, St Louis, MO, USA) and centrifuged at 14,000g for 10 min at 4°C. Proteins in the supernatant were denatured and then incubated with PNGase F or Endo H (New England Biolabs, Ipswich, MA, USA) following the instructions of the manufacturer. Deglycosylation reactions were performed for 30 min and stopped by adding SDS-PAGE loading buffer and boiling. Proteins were detected by Western blot using anti-HA antibodies as described below.

## Concanavalin A purification

Proteins were extracted in 50 mM Tris-HCl, pH 7.5, 150 mM NaCl, 10% [v/v] glycerol, 1 mM PMSF, and 1% plant protease inhibitor cocktail (Sigma-Aldrich) and centrifuged at 14,000 g for 10 min at 4°C. The supernatant was equilibrated in concanavalin A buffer (0.2 M Tris-HCl pH 7.5, 1 M NaCl, 200 mM $MgCl_2$, 200 mM $CaCl_2$) and applied to concanavalin A-agarose resin from *Canavalia ensiformis* (Sigma-Aldrich) pre-equilibrated in concanavalin A buffer. After three steps of washing with

concanavalin A buffer, glycosylated proteins were eluted in concanavalin A buffer supplemented with 0.75 M α-D-methyl-glucoside and 0.75 M α-D-methylmannoside. The presence of HA-tagged SBT5.2 in the eluted proteins was confirmed by Western blot using anti HA antibodies as described below.

### Isolation of intercellular (apoplastic) fluid

*N. benthamiana* leaves transiently expressing the proteins of interest were harvested 48 hr after agroinfiltration and infiltrated with water. Intercellular fluids (IF) were isolated by centrifugation at 3000 g as previously described (*de Wit and Spikman, 1982*).

### Gel blot analysis

Antibodies used for Western blotting were anti-HA-HRP (3F10, Roche, Germany, 1:5000), and PAP anti-rabbit-HRP (Sigma, 1:10,000). Proteins were visualized using the Immobilon kit (Millipore, Billerica, MA, USA) under standard conditions.

### Transient transfection of Arabidopsis protoplasts

The isolation and transient transfection of leaf mesophyll cell protoplasts from *Arabidopsis* plants (four weeks-old) was performed at room temperature following published procedures (*Yoo et al., 2007*). A total of 10 µg plasmid DNA was used for each transfection experiment and plasmids were mixed in an equal ratio for cotransfections.

### Fluorescein isothiocyanate(FITC)-labeled casein assay for proteolytic activity

FITC casein was used for the detection of proteolytic activity (*Twining, 1984*). *sbt5.2* Arabidopsis protoplasts transfected with SBT5.2(a), SBT5.2(a)$^{H210A}$, SBT5.2(b), SBT5.2(b)$^{H171A}$ or empty vector were lysed and used for the activity assay. A concentration of 400 µg/ml of FITC casein was used in a final volume of 25 µl of 150 mM NaCl, 20 mM phosphate buffer pH 7.6. Samples were incubated for 1 hr at 25°C and fluorescence was measured at excitation 485 nm and emission 520 nm using a microtiter fluorimeter (FL600, Bio-Tek, Highland Park, VT, USA).

### Quantification of cell death

For electrolyte leakage measurement, four leaf discs (6-mm diameter) were harvested at the indicated timepoints after plant inoculation, washed, and incubated at room temperature in 5 ml of distilled water before measuring conductivity. The production of phenolic compounds was monitored under UV light using a Zeiss Axioplan microscope 24 hr after inoculation.

### RNA extraction and Q-RT-PCR analysis

Material for RNA analysis was grounded in liquid nitrogen and total RNA was isolated using the Nucleospin RNA plant kit (Macherey-Nagel, Germany) according to the manufacturer's recommendations. Reverse transcription was performed using 1.5 µg of total RNA. Real-time quantitative PCR was performed on a Light Cycler 480 II machine (Roche Diagnostics, France), using Roche reagents. Primers used for Q-RT-PCR are provided as Supporting Information. Relative expression was calculated as the ΔCp between each gene and the internal controls *SAND* family (At2g28390). Average ΔCp was related to the value of each gene in each line at time 0.

## Acknowledgements

We thank Céline Remblière for help with plant transformation. We thank Roger Innes for kindly providing the multisite gateway system, as well as VHA-a1 and ARA6 cDNAs and GmMan1 construct. We are grateful to Paul Hurst for help with plant inoculation and conductivity assays. PB was funded by a grant from the French Ministry of National Education and Research. IS is supported by INRA (Institut National de la Recherche Agronomique) and an AgreenSkills fellowship within the EU Marie-Curie FP7 COFUND People Programme (grant agreement n° 267196). Our work is supported by the French Laboratory of Excellence project 'TULIP' (ANR-10-LABX-41; ANR-11-IDEX-0002-02).

## Additional information

### Funding

| Funder | Grant reference number | Author |
| --- | --- | --- |
| Institut National de la Recherche Agronomique (INRA) | | Irene Serrano |
| AgreenSkills fellowship, EU Marie-Curie FP7 COFUND People Programme | grant agreement n° 267196 | Irene Serrano |
| Freach Ministry of National Education and Research | | Pierre Buscaill |
| French Laboratory of Excellence project 'TULIP' | ANR-10-LABX-41 | Susana Rivas |
| French Laboratory of Excellence project 'TULIP' | ANR-11-IDEX-0002-02 | Susana Rivas |

The funders had no role in study design, data collection and interpretation, or the decision to submit the work for publication.

### Author contributions

IS, SR, Conception and design, Acquisition of data, Analysis and interpretation of data, Drafting or revising the article; PB, CA, CP, AJ, Acquisition of data, Analysis and interpretation of data

### Author ORCIDs

Susana Rivas, http://orcid.org/0000-0002-2549-7346

## Additional files

### Supplementary files

• Supplementary file 1. Oligonucleotide primers used in this study.

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
