## [Decision Letter]

Thank you for submitting your article "A non canonical subtilase attenuates the transcriptional activation of defence responses in *Arabidopsis thaliana*" for consideration by *eLife*. Your article has been favorably evaluated by Detlef Weigel as the Senior Editor and three reviewers: Andreas Schaller (Reviewer #2), Marty Dickman (Reviewer #3), and a member of our Board of Reviewing Editors.

The reviewers have discussed the reviews with one another and the Reviewing Editor has drafted this decision to help you prepare a revised submission.

Rivas and coworkers show that the subtilisin-like protease SBT5.2 is a negative regulator of defense in *Arabidopsis* and that it acts through repression of MYB30 transcriptional activity. They further show that there are two splice variants, SBT5.2(a) which represents the canonical secreted subtilase, and SBT5.2(b) which lacks a signal peptide and thus fails to be secreted. SBT5.2(b) is targeted to enodosomes by N-terminal myristoylation. It interacts with MYB30 via its C-terminal domain, sequestering the transcription factor in the cytosol thus preventing nuclear localization and activation of defense. These findings are novel and highly relevant. They provide important insight into the function of subtilases and reveal a novel mechanism of transcriptional regulation.

Summary:

This is a thorough and comprehensive study. The paper is well written and the work is rigorously performed. Most of the conclusions are supported by multiple lines of evidence. Experiments are generally well-controlled and data appear solid.

Essential revisions:

Note that the reviewers did not find major flaws with the paper but did raise a number of more minor comments that the authors will need to address. These are listed below.

Please clarify the following, which the reviewers found either confusing or misleading:

1) The prodomain serves as an intramolecular inhibitor and keeps SBTs inactive until it is cleaved and released from the enzyme. Because SBT5.2(b) still has its prodomain, I would expect it to be inactive. This should be clarified in the text in case not all readers are specialists in this area.

2) There was a general concern about studies of protein activity based strictly on crude extracts that might, for example, contain inhibitors that could modulate activity, separate from that intrinsic to a purified protein. The authors need to address this concern and, at least, provide some caveats for their claims based on these experiments.

3) Continuing with this concern, with respect to the suitability of the activity assay, the reviews raised concerns about the use of crude extracts on a universal protease substrate, resulting in high background activity. There is an increase of only 10% in protoplasts expressing SBT5.2(a). There were doubts about whether this assay is really suitable for extracellular proteins. Secreted proteins would likely be diluted in the medium and lost when lysed protoplasts are used for the activity assay.

4) Is expression of SBT5.2(b) upregulated in the *myb30* mutant background? The authors seem to imply that this gene is under MYB30 control.

5) The authors need to confirm that their prediction of higher MYB30 levels in the *sbt5.2* mutant plants is indeed occurring.

6) It is known that only the processed form of SBTs is secreted, the precursor is not. Therefore, only the processed form is expected to be present in intercellular fluids (IF). However, in Figure 1 the SBT5.2(a) precursor is clearly detected in IF, and the ratio of band intensities for precursor and mature SBT5.2(a) appears to be the same in TE and in IF. This would argue for contamination with intracellular proteins.

7) In Figure 2, deglycosylation resulted in a mobility shift for both, the precursor and the mature protease. This observation raises doubts whether SBT5.2(a) is actually secreted, because during passage through the secretory pathway, the cleavable high-mannose precursors of N-linked glycans are modified. Plant-specific modifications include α(1,3)-fucosylation and β 1,2-xylosylation of the glycan core which renders them resistant to cleavage by EndoH and PNGase F (ref 1). Therefore, deglycosylation and the resulting mobility shift are only expected for the precursor that did pass through the Golgi yet. The mature protease, if secreted, would be expected to be EndoH and PNGase F resistant.

8) As an easy experiment to confirm secretion of the protease would suggest to include plasmolysis treatment in the transient expression experiment shown in Figure 1.

---

## [Author Response]

*Essential revisions:*

*1) The prodomain serves as an intramolecular inhibitor and keeps SBTs inactive until it is cleaved and released from the enzyme. Because SBT5.2(b) still has its prodomain, I would expect it to be inactive. This should be clarified in the text in case not all readers are specialists in this area.*

The notion of the prodomain as an intramolecular inhibitor of protease activity, which was previously discussed in the third paragraph of the Discussion, is now introduced earlier in the first section of Results.

*2) There was a general concern about studies of protein activity based strictly on crude extracts that might, for example, contain inhibitors that could modulate activity, separate from that intrinsic to a purified protein. The authors need to address this concern and, at least, provide some caveats for their claims based on these experiments.*

Our first attempts to test SBT5.2(a) and (b) activity were indeed performed with purified proteins following immunoprecipitation from crude extracts. This protocol confirmed detection of increased fluorescence (indicative of protease activity) for SBT5.2(a), but not SBT5.2(a)^H210A^, SBT5.2(b), SBT5.2(b)^H171A^, as compared to empty vector. However, we failed to detect SBT5.2(b) proteins by Western blot after immunoprecipitation. Indeed, SBT5.2(b) is highly prone to degradation, likely due to lack of glycosylation (glycosylation has been described to play a role in protein stabilization and protection from degradation in both plant and animal cells). In contrast, glycosylated SBT5.2(a) is more stable both after protein extraction and in vivo (see Figure 2). Although we modified the incubation time and temperature to try and reduce SBT5.2(b) degradation, we did not succeed in detecting SBT5.2(b) protein (or activity) after immunoprecipitation. Under these conditions, it was thus not possible to conclude whether lack of increased fluorescence was due to lack of SBT5.2(b) protease activity or just to the fact that the protein was absent owing to degradation during immunoprecipitation. We additionally tried to express and purify the proteins in bacteria but the subtilases were systematically found in inclusion bodies, which was not compatible with activity assays.

However, it is important to note that the goal of these assays was not to show the activity of SBT5.2(a), as this was previously demonstrated in two independent previous reports (Kaschani et al., 2012; Engineer et al., 2014), but to place ourselves in the best possible conditions to detect activity of SBT5.2(b). We therefore worked with total extracts which allowed the best preservation of the protein. However, even in these favorable conditions and in agreement with other biochemical evidences presented in the article, we were not able to detect SBT5.2(b) protease activity.

*3) Continuing with this concern, with respect to the suitability of the activity assay, the reviews raised concerns about the use of crude extracts on a universal protease substrate, resulting in high background activity. There is an increase of only 10% in protoplasts expressing SBT5.2(a). There were doubts about whether this assay is really suitable for extracellular proteins. Secreted proteins would likely be diluted in the medium and lost when lysed protoplasts are used for the activity assay.*

We agree with the reviewer that the secreted protein is probably lost in the medium and not detectable in this assay. Therefore, the detected activity (and protein in the Western blot) most likely corresponds to the mature (active) SBT5.2(a) form that is about to be secreted but still intracellular. Although it is not possible to know the proportion of intracellular versus secreted SBT52(a), it appears to be high enough for detection of its activity. Indeed, 14% is not a high increase of activity as compared to background levels but SBT5.2(a) fluorescence levels were consistently found to be higher than those detected in SBT5.2(a)^H210A^, SBT5.2(b), SBT5.2(b)^H171A^ or empty vector-expressing protoplasts, which confirms protease activity.

In any case, loss of secreted proteins in the extracellular medium should not be a problem for detection of activity of intracellular proteases such as SBT5.2(b), which as mentioned in the point above was the goal of these assays. We thus consider that this assay confirms lack of SBT5.2(b) activity.

*4) Is expression of SBT5.2(b) upregulated in the myb30 mutant background? The authors seem to imply that this gene is under MYB30 control.*

We apologize if our text conveyed this idea. Indeed, we do not mean to imply that SBT5.2(b) expression is controlled by MYB30 but that expression of both genes follows a similar expressing pattern after bacterial inoculation. We have modified a sentence at the end of the Introduction to clarify this point.

*5) The authors need to confirm that their prediction of higher MYB30 levels in the sbt5.2 mutant plants is indeed occurring.*

We are not sure to understand the reviewer’s comment. We do not expect higher MYB30 levels in the sbt5.2 mutant plants, as the regulation of MYB30 by SBT5.2(b) is independent of SBT5.2(b) catalytic activity and has no effect on MYB30 gene expression or protein accumulation levels but on its subcellular localization. By retaining MYB30 in endosomal vesicles, SBT5.2(b) impedes MYB30 nuclear entry and activation of its target genes.

*6) It is known that only the processed form of SBTs is secreted, the precursor is not. Therefore, only the processed form is expected to be present in intercellular fluids (IF). However, in Figure 1 the SBT5.2(a) precursor is clearly detected in IF, and the ratio of band intensities for precursor and mature SBT5.2(a) appears to be the same in TE and in IF. This would argue for contamination with intracellular proteins.*

We agree with the reviewer that this is a surprising observation, since the general trend is the presence of the processed form in the IF. We do not think that this is due to contamination with intracellular proteins since we did not detect intracellular MIEL1 in these assays and systematically detected the slow migrating SBT5.2(a) band that we think may correspond to the unprocessed form of the protein. In addition, as you can see in Figure 10, the catalytic mutant SBT5.2(a)-H210A was also detected in the IF although this protein is unable to mediate its selfprocessing and is also slow migrating. Understanding this observation requires further investigation but is beyond the scope of this work, which is focused on SBT5.2(b)-mediated regulation of MYB30-dependent plant defence.

Author response image 1.Western blot analysis of total protein extracts (TE) and intercellular fluids (IF) from *N. benthamiana* leaves co-expressing HA-tagged intracellular MIEL1 with wild-type and catalytically mutant SBT5.2(**a**) proteins.**DOI:**
http://dx.doi.org/10.7554/eLife.19755.019

*7) In Figure 2, deglycosylation resulted in a mobility shift for both, the precursor and the mature protease. This observation raises doubts whether SBT5.2(a) is actually secreted, because during passage through the secretory pathway, the cleavable high-mannose precursors of N-linked glycans are modified. Plant-specific modifications include α(1,3)-fucosylation and β 1,2-xylosylation of the glycan core which renders them resistant to cleavage by EndoH and PNGase F (ref 1). Therefore, deglycosylation and the resulting mobility shift are only expected for the precursor that did pass through the Golgi yet. The mature protease, if secreted, would be expected to be EndoH and PNGase F resistant.*

We are sorry that we were not clear enough about this point in the previous version of the manuscript. We did not mean to convey the idea that deglycosylase treatment results in a mobility shift for both the processed and the unprocessed form of SBT5.2(a). In agreement with the reviewer’s comment, the mobility of the unprocessed band is increased after deglycosylation whereas the mobility of the processed form (that is expected to be resistant to EndoHand PNGaseF treatment) is not affected by deglycosylase treatment as expected. In order to fully clarify this point and better illustrate this idea, we have repeated the experiment using a longer time to run the gel and thus improve separation of the different bands. Please see new Figure 2 in which the fully processed form is indicated with an arrowhead.

8) As an easy experiment to confirm secretion of the protease would suggest to include plasmolysis treatment in the transient expression experiment shown in Figure 1.

In our view secretion of SBT5.2(a) has been clearly demonstrated. First, as mentioned in the text, SBT5.2(a) was first identified in an activity-based probe assay to display the activity of apoplastic serine hydrolases (Kaschani et al., 2012). Second, a cell-wall mass spectrometry analysis identified SBT5.2(a) as a CO_2_-induced extracellular protease (Engineer et al., 2014). Third, Figure 1 clearly shows accumulation of SBT5.2(a)-RFP in apoplastic spaces. To make this clearer for the reader, apoplastic spaces where SBT5.2(a)-RFP accumulates are indicated with arrowheads in the new version of Figure 1. Fourth, further confirmation of the extracellular accumulation of SBT5.2(a) was obtained by isolation of IF (Figure 1). Based on all these results, we do not consider it necessary to include an additional figure with a plasmolysis treatment.